# COMBATING THE GENERALIZATION-FORGETTING TRADE-OFF IN CONTINUAL LEARNING: A CAUTIOUS PASSIVE LOW-RANK APPROACH

## ABSTRACT

Large Language Models (LLMs) have shown remarkable capabilities through wide-scale pre-training on a wide range of domains. However, they often suffer from catastrophic forgetting when learning sequential tasks. In this paper, we propose a novel parameter-efficient approach for continual learning in LLMs, which empirically explores the role of different effective layerwise ranks, leveraging lower ranks to mitigate catastrophic forgetting of previous tasks and higher ranks to enhance generalization on new tasks. By employing a subspace similarity metric that evaluates the orthogonality of low-rank subspaces between tasks, we gradually increase the rank of layerwise matrices for each new task, minimizing interference with previously learned tasks while enhancing generalization. Experimental results on standard continual learning benchmarks and challenging math benchmarks demonstrate that our method outperforms existing state-of-the-art approaches, effectively mitigating forgetting, improving task performance, and maintaining strong generalization to unseen tasks in a memory-efficient manner.

## 1 INTRODUCTION

As Large Language Models (LLMs) (Raffel et al., 2020; Chowdhery et al., 2023; Achiam et al., 2023; Touvron et al., 2023) continue to scale, adapting pre-trained foundation models to numerous downstream tasks become common practice, but fully fine-tuning these models is impractical given the large model sizes. Consequently, low-rank adaptation methods like LoRA Hu et al. (2021) and its multiple variants (Zhang et al., 2023b; Liu et al., 2024) have emerged to enable parameter-efficient fine-tuning for LLMs.

While pre-trained LLMs have achieved great success on fine-tuning on static tasks, continual learning (CL), the process of learning multiple sequential tasks, remains a significant challenge Wu et al. (2021; 2024). Two key obstacles are (i) catastrophic forgetting, where a model's performance on earlier tasks degrades when trained on new tasks (McCloskey & Cohen, 1989; Ratcliff, 1990), and (ii) generalization ability, where the previously learned model improves new tasks. Within the realm of LLMs, CL goes beyond enhancing linguistic and reasoning abilities, involving complex processes such as continual pretraining Jin et al. (2021), continual instruction Zhang et al. (2023c), and continual alignment Zhang et al. (2023a).

Although existing LoRA-based parameter-efficient tuning (PET) methods for CL have mitigated the forgetting issue, such as O-LoRA Wang et al. (2023) that incrementally learns new tasks in orthogonal subspaces, most approaches apply the same rank across all layers in the model. However, the effectiveness of heterogeneous nature of different layers in overparameterized models has been extensively studied as highlighted in Zhang et al. (2022). Moreover, in the context of pre-training and adaptation for LLMs, AdaRank Dong (2024) introduces a simple model disagreement-based technique for determining layerwise ranks for low-rank adaptation induced by random module perturbations. Additionally, both AdaLoRA Zhang et al. (2023b) and SoRA Ding et al. (2023) exploit the relationship between the rank and the singular value decomposition of the weight update matrices to dynamically adjust layerwise ranks during adaptation. Specifically, AdaLoRA achieves this by pruning the singular values associated with less significant updates, while SoRA employs a learnable gating mechanism that gradually reduces the rank as training progresses. These findings

strongly suggest that using different ranks for different layers is more effective, as enforcing the same rank across all layers may lead to overfitting certain features and diminished generalizability.

Several studies have shown that LoRA forgets less than common regularization techniques like weight decay and dropout (Biderman et al., 2024; Hyder et al., 2022), and LoRA helps maintain the diversity of generations. The results in Biderman et al. (2024) show that LoRA forgets less than full fine-tuning. However, the low-rank update mechanism limits the ability of LLMs to learn and retain new knowledge as effectively as full fine-tuning (Hu et al., 2021; Xia et al., 2024; Hao et al., 2024; Zhao et al., 2024), especially in challenging tasks like mathematical reasoning. COLA addresses this by employing an iterative low-rank residual learning process to approximate the optimal weight updates for task adaptation Xia et al. (2024), somewhat increasing the ranks of LoRAs by extending the chain length. FLORA achieves high-rank updates by resampling the projection matrices to mitigate the low-rank limitation of LoRA Hao et al. (2024). While rank dynamics have been explored in the context of static fine-tuning tasks, to our knowledge, no study in CL for LLMs has thoroughly examined these rank patterns.

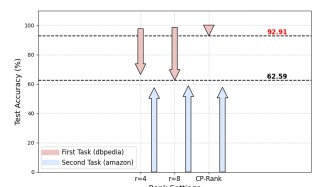

(a) Each task testing accuracy after training task $\mathcal{T}_2$ (where $\mathcal{T}_1$: dbpedia, $\mathcal{T}_2$: amazon).

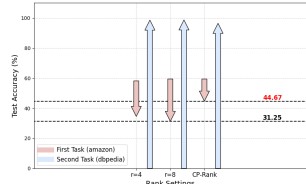

(b) Each task testing accuracy after training task $\mathcal{T}_2$ (where $\mathcal{T}_1$: amazon, $\mathcal{T}_2$: dbpedia).

Figure 1: Comparison of each task accuracy changes after training task $\mathcal{T}_2$ in two different task orders. Although ranks 4 and 8 achieve similar average accuracy, rank 8 causes greater accuracy loss on $\mathcal{T}_1$ after training $\mathcal{T}_2$ while rank 4 fails to match the performance of rank 8 on both $\mathcal{T}_1$ and $\mathcal{T}_2$.

To examine the impact of layerwise ranks in incremental learning of LoRA between tasks for CL, we conduct an experiment using a fixed uniform rank across all layers, testing two different rank settings and freezing previously learned incremental LoRAs without regularization when training new tasks. The results, shown in Fig. 1 using pre-trained T5-large model Raffel et al. (2020) with fixed-rank incremental LoRAs for DBpedia and Amazon Reviews Zhang et al. (2015), indicate that while similar average accuracy across tasks can be achieved with both low and high ranks in certain cases, higher ranks for the second task tend to cause greater accuracy loss on the first task, and lower ranks for both tasks cannot achieve the same good performance on the current task as higher ranks. Importantly, this does not imply a straightforward linear relationship between rank size and reduced forgetting. Instead, it reveals a trade-off between low ranks and high ranks to balance forgetting mitigation and generalization. These observations highlight the need for an ideal approach in CL for LLMs within the PET framework, one that utilizes the role of the layerwise tanks to balance catastrophic forgetting and generalization across a continual stream of tasks. Inspired by this, we aim to address the following fundamental question:

*How can we design an adaptive parameter-efficient CL algorithm that leverages the **forgetting**-mitigation nature of low ranks and the **generalization** strengths of high ranks to optimize the trade-off?*

To answer this question, we propose a novel adaptive algorithm dubbed as CP-Rank (**C**autious **P**assive Low-**Rank**), that *gradually increases the rank of layerwise weight matrices* during training among layers. This is accomplished by empirically examining how layerwise ranks affect both forgetting and generalization, with rank adjustments guided by a between-task low-rank subspace similarity metric. Specifically, CP-Rank focuses on the subsequent tasks after the first task. For each task after the first task, CP-Rank starts by setting the incremental LoRA rank to 1, aiming to minimize interference with previously learned tasks from the beginning of new task training. It then applies SVD decomposition to compute the left singular layerwise matrices of LoRAs from both the current and previous tasks, thus calculating the subspace similarity between their low-rank matrices. With this dynamic similarity during training, CP-Rank evaluates the orthogonality of the subspaces and decides whether to **cautiously increase** the rank for the current task by allocating additional low-rank parameters, or whether to **passively maintain** the current rank, balancing learning of new information with retention of previously acquired knowledge. It is important to note that CP-Rank freezes all previously learned incremental LoRAs during the training of each new task. Our experimental results demonstrate that CP-Rank outperforms state-of-the-art methods on standard continual learning benchmarks and excels in more challenging math tasks, such as GSM8K Cobbe et al. (2021) and MATH Hendrycks et al. (2021). Furthermore, our analysis explores the impact of various hyperparameters and evaluates different rank update rules, highlighting CP-Rank's effectiveness in robustness to task orders, mitigating forgetting, and enhancing generalization.

■ **Summary of Contributions.** This paper makes three key contributions: (1) A novel parameter-efficient continual learning method for LLMs that effectively balances forgetting and generalization through cautious passive low-rank updates; (2) Through comprehensive evaluations, our method demonstrates superior performance over existing state-of-the-art approaches both on standard continual learning benchmarks and math datasets; and (3) We provide an in-depth analysis that deepens our understanding of the dynamics of gradually increasing rank within continual learning for LLMs, pinpointing critical factors that drive its effectiveness.

## 2 CAUTIOUS PASSIVE LOW-RANK CONTINUAL LEARNER

In this section, we propose a parameter-efficient continual learning approach that cautiously increases the rank and passively maintains the rank during training, leveraging low ranks to reduce forgetting and high ranks to improve generalization.

■ **Problem Setting.** In the continual learning scenario, we have a sequence of tasks $\mathcal{T} = \{\mathcal{T}_1, \mathcal{T}_2, \ldots, \mathcal{T}_N\}$ over time. Each task $\mathcal{T}_k$ is associated with a data distribution $\mathcal{D}_k$ and contains a separate target dataset $\mathcal{S}_k = \{(\boldsymbol{x}_{k,i}, y_{k,i})\}_{i=1}^{n_k}$ where $\boldsymbol{x}_{k,i} \in \mathcal{X}_k$ and $y_{k,i} \in \mathcal{Y}_k$. The goal of continual learning is to find a set of parameters $\boldsymbol{\theta} \in \Theta$ that can effectively solve all tasks up to the current task $\mathcal{T}_k$, while minimizing catastrophic forgetting of previously learned tasks. In continual learning of LLMs, we are given a pre-trained model $\boldsymbol{W}_0$ and would like to continually fine-tune a sequence of tasks, utilizing the incremental low-rank matrix parameters $\boldsymbol{B}_k \boldsymbol{A}_k$ to finetune task $\mathcal{T}_k$ where $\boldsymbol{B}_k \in \mathbb{R}^{d_1 \times r}, \boldsymbol{A}_k \in \mathbb{R}^{r \times d_2}$ and the rank $r \ll \min(d_1, d_2)$. The continual learning model parameters after fine-tuning on task $\mathcal{T}_k$ is $\boldsymbol{\theta}_k = \boldsymbol{W}_0 + \sum_{s=1}^{k} \boldsymbol{B}_s \boldsymbol{A}_s$. Our continual learning goal is to optimize the following objective across all tasks:

$$\max_{\boldsymbol{\theta}} \sum_{k=1}^{N} \sum_{(\boldsymbol{x},y) \in \mathcal{S}_k} \log p_{\boldsymbol{\theta}}(y|\boldsymbol{x}), \tag{1}$$

where $\boldsymbol{\theta} = \boldsymbol{W}_0 + \sum_{k=1}^{N} \boldsymbol{B}_k \boldsymbol{A}_k$. It is important to note that in our scenario, the model does not have access to data from previous tasks when learning a new task, while the model predicts sample labels without knowledge of the corresponding task ID.

■ **Forgetting Error Bound in Low-Rank CL.** The forgetting error in CL, which measures the degradation in performance on previously learned tasks after learning a new task is formulated as:

$$\mathcal{F}(\boldsymbol{\theta}_1, \ldots, \boldsymbol{\theta}_N) = \sum_{t=1}^{N-1} \mathcal{L}_t(\boldsymbol{\theta}_T) - \mathcal{L}_t(\boldsymbol{\theta}_t) \tag{2}$$

where $\boldsymbol{\theta}_t = \boldsymbol{W}_0 + \sum_{k=1}^{t} \boldsymbol{B}_k \boldsymbol{A}_k$, $\mathcal{L}_t(\cdot)$ is the generalization error on task $\mathcal{T}_t$, and $\mathcal{L}_t(\boldsymbol{\theta}_T) - \mathcal{L}_t(\boldsymbol{\theta}_t)$ is the performance degradation (forgetting) on tasks $\mathcal{T}_t$ between the model after training on task $\mathcal{T}_t$ and the model after training on the final task $\mathcal{T}_N$. The generalization error, which assesses the model capability to effectively learn a new task while preserving the knowledge acquired from previous tasks is defined as:

$$\mathcal{I}(\boldsymbol{\theta}_1, \ldots, \boldsymbol{\theta}_N) = \sum_{t=1}^{N} \mathcal{L}_t(\boldsymbol{\theta}_t) - \mathcal{L}_t(\boldsymbol{\theta}_t^*) \tag{3}$$

where $\mathcal{L}_t(\boldsymbol{\theta}_t) - \mathcal{L}_t(\boldsymbol{\theta}_t^*)$ measures the generalization gap between the CL model $\boldsymbol{\theta}_t$ and the optimally fine-tuned model $\boldsymbol{\theta}_t^* = \boldsymbol{W}_0 + \boldsymbol{B}_t^* \boldsymbol{A}_t^*$ on task $\mathcal{T}_t$. The generalization of final model on all tasks can be decomposed into forgetting-generalization errors as follows:

$$\sum_{t=1}^{N} \mathcal{L}_t(\boldsymbol{\theta}_N) - \mathcal{L}_t(\boldsymbol{\theta}_t^*) = \mathcal{F}(\boldsymbol{\theta}_1, \ldots, \boldsymbol{\theta}_N) + \mathcal{I}(\boldsymbol{\theta}_1, \ldots, \boldsymbol{\theta}_N) \tag{4}$$

To provide intuition about the proposed algorithm, we start by examining the forgetting error in a simple linear regression setting with $N = 2$ and $n_1 = n_2 = n$ (for detailed derivation, please see Appendix A.5). While a larger rank is preferable to entail a better generalization on a new task, the effect of rank on forgetting highly depends on similarity between tasks which can be bounded by:

$$\mathbb{E}[\mathcal{F}(\boldsymbol{\theta}_1, \boldsymbol{\theta}_2)] \lesssim \mathcal{O}(\text{tr}((\boldsymbol{B}_1 \boldsymbol{A}_1)(\boldsymbol{B}_2 \boldsymbol{A}_2)^\top) + \text{additional terms.} \tag{5}$$

This finding motivates us to find an effective subspace similarity between tasks during training to control and optimize the forgetting and generalization trade-off.

■ **Between-Task Different-Rank Layerwise Subspace Similarity Measure.** To measure the low-rank subspace similarity between different tasks, we utilize a reverse metric Hu et al. (2021) of the standard Projection Metric of Grassmann Distance that measures the distance between subspaces Hamm & Lee (2008). For any two tasks, we define the low-rank subspace at layer $l$ of task

---

**Algorithm 1:** Cautious Passive Low-Rank Continual Learning for Task $\mathcal{T}_i, i \in [2, N]$

---

**Require:** Starting rank $r_i^0 = 1$, interval $k \in \mathbb{Z}_+$, total updating steps $T$

1 Initialize $\boldsymbol{A}_i^0 \in \mathbb{R}^{r_i^0 \times d_2}$ using random Gaussian initialization and $\boldsymbol{B}_i^0 \in \mathbb{R}^{d_1 \times r_i^0}$ as zero initialization

2 $t \leftarrow 1$

3 **while** $t < T$ **do**

4    **if** $t \equiv 0 \mod k$ **then**

5       | Obtain $\boldsymbol{A}_i^t$ and $\boldsymbol{B}_i^t$ from Algorithm 2

6    **end**

7    Train low-rank network and obtain $\boldsymbol{A}_i^{t+1}$ and $\boldsymbol{B}_i^{t+1}$

8    $t \leftarrow t + 1$

9 **end**

---

$\mathcal{T}_i$ as $\boldsymbol{B}_i^l \boldsymbol{A}_i^l$, where $\boldsymbol{B}_i^l \in \mathbb{R}^{d_1 \times r_i}$ and $\boldsymbol{A}_i^l \in \mathbb{R}^{r_i \times d_2}$. Similarly, the low-rank subspace at layer $l$ of task $\mathcal{T}_j$ is defined as $\boldsymbol{B}_j^l \boldsymbol{A}_j^l$, where $\boldsymbol{B}_j^l \in \mathbb{R}^{d_1 \times r_j}$ and $\boldsymbol{A}_j^l \in \mathbb{R}^{r_j \times d_2}$. We first perform SVD decomposition on the low-rank subspaces of tasks $\mathcal{T}_i$ and $\mathcal{T}_j$ to obtain their respective top $r_i$ and top $r_j$ left singular vectors: $\boldsymbol{U}_i^l \in \mathbb{R}^{d_1 \times r_i}$ and $\boldsymbol{U}_j^l \in \mathbb{R}^{d_1 \times r_j}$. Then we let the singular values of $(\boldsymbol{U}_i^l)^\top \boldsymbol{U}_j^l$ to be $\sigma_1, \sigma_2, \ldots, \sigma_p$, where $p = \min\{r_i, r_j\}$. The Grassmann Distance standard projection metric is defined as:

$$d(\boldsymbol{U}_i^l, \boldsymbol{U}_j^l) = \sqrt{p - \sum_{s=1}^{p} \sigma_s^2} \in [0, \sqrt{p}] \tag{6}$$

Following LoRA Hu et al. (2022) and the Grassmann Distance, we define our task subspace similarity metric as:

$$\phi(\boldsymbol{U}_i^l, \boldsymbol{U}_j^l) = \frac{\sum_{i=s}^{p} \sigma_s^2}{p} = \frac{1}{p}\left(1 - d(\boldsymbol{U}_i^l, \boldsymbol{U}_j^l)^2\right) \tag{7}$$

This similarity metric satisfies the following conditions: when $\boldsymbol{U}_i^l$ and $\boldsymbol{U}_j^l$ share the same column span, considered as overlapping, then $\phi(\boldsymbol{U}_i^l, \boldsymbol{U}_j^l) = 1$. If they are completely orthogonal, then $\phi(\boldsymbol{U}_i^l, \boldsymbol{U}_j^l) = 0$. Otherwise, $\phi(\boldsymbol{U}_i^l, \boldsymbol{U}_j^l) \in (0, 1)$. We use this metric to determine whether the low-rank subspaces of two tasks are orthogonal. If $\phi(\boldsymbol{U}_i^l, \boldsymbol{U}_j^l) < \epsilon$, we consider the low-rank subspaces of task $\mathcal{T}_i$ and task $\mathcal{T}_j$ are orthogonal, meaning increasing the rank in the current subspace is "safe" for both tasks, as it would not interfere with the learned low-rank subspaces of previous tasks. Conversely, if $\phi(\boldsymbol{U}_i^l, \boldsymbol{U}_j^l) > \epsilon$, the subspaces are not orthogonal, and we maintain the current rank for the new task to reduce the risk of forgetting prior tasks. Moreover, Eq. 7 uses the singular values captured by two different task subspaces, which matches our findings in Eq. 5.

■ **Cautious Passive Low-Rank Continual Learning.** We now turn to providing the detailed algorithm. For simplicity, we use $\boldsymbol{B}$ and $\boldsymbol{A}$ to represent the layer-wise weight matrices $\boldsymbol{B}^l$ and $\boldsymbol{A}^l$ at layer $l$.

**For task $\mathcal{T}_1$.** In our method, we focus primarily on the subsequent tasks after the first task, as the subspace similarity metric is designed to evaluate low-rank weight subspaces between tasks. Since the first task has no previous tasks to compare against, we use a fixed low rank $\boldsymbol{B}_1 \boldsymbol{A}_1$ for learning.

**For task $\mathcal{T}_i, i \in [2, N]$.** When training task $\mathcal{T}_i$, we freeze the low-rank matrices of all previous tasks $\{\mathcal{T}_m\}_{m=1}^{i-1}$. For task $\mathcal{T}_i$, CP-Rank initializes the low-rank matrices $\boldsymbol{B}_i \boldsymbol{A}_i$ with a rank of 1, minimizing the impact on previously learned tasks, as done in other incremental low-rank methods Zhao et al. (2023). Next, we perform SVD on the low-rank matrix $\boldsymbol{B}_i \boldsymbol{A}_i$ to obtain the top $r_i$ left singular vectors $\boldsymbol{U}_i$. Similarly, we compute the top $r_m$ left singular matrices $\{\boldsymbol{U}_m\}_{m=1}^{i-1}$ for the previous tasks $\{\mathcal{T}_m\}_{m=1}^{i-1}$ low-rank matrices $\{\boldsymbol{B}_m \boldsymbol{A}_m\}_{m=1}^{i-1}$. Using these matrices, we calculate the subspace similarity $\phi(\boldsymbol{U}_i, \boldsymbol{U}_m)$ between the subspaces of the current task $\mathcal{T}_i$ and each previous task $\mathcal{T}_m$ to obtain the average subspace similarity. Based on this average subspace similarity, if it's below the orthogonality threshold, indicating that $\boldsymbol{U}_i$ is sufficiently orthogonal to the previous ones, CP-Rank cautiously increases the rank of $\boldsymbol{B}_i$ and $\boldsymbol{A}_i$ to improve generalization without negatively impacting earlier tasks. Otherwise, the rank passively remains unchanged to avoid interference with previous tasks. The complete algorithm is outlined in Algorithms 1 and 2.

---

**Algorithm 2:** Cautious Passive Low Rank Update

---

**1** **for** $m \le i - 1$ **do**
**2**     Compute left singular matrix: $\boldsymbol{U}_m \leftarrow \mathrm{SVD}(\boldsymbol{B}_m \boldsymbol{A}_m)$ of task $\mathcal{T}_m$
**3**     Select top $r_m$ left singular vectors of $\boldsymbol{U}_m$
**4** **end**
**5** Compute and obtain top $r_i$ left singular vectors from task $\mathcal{T}_i$: $\boldsymbol{U}_i^t \leftarrow \mathrm{SVD}(\boldsymbol{B}_i^t \boldsymbol{A}_i^t)$
**6** Compute task subspace similarity $\phi(\boldsymbol{U}_i^t, \boldsymbol{U}_m)$, where $m = 1, \ldots, i - 1$
**7** **if** $\frac{1}{i-1} \sum_{m=1}^{i-1} \phi(\boldsymbol{U}_i^t, \boldsymbol{U}_m) < \epsilon$ **then**
**8**     $r_i^t = r_i^t + 1$
**9**     Initialize additional parameters: $\boldsymbol{A}_i^t \leftarrow [\boldsymbol{A}_i^t, \boldsymbol{A}^*], \boldsymbol{B}_i^t \leftarrow [\boldsymbol{B}_i^t, \boldsymbol{B}^*]$, where $\boldsymbol{A}^* \in \mathbb{R}^{1 \times d_2}$ and
      $\boldsymbol{B}^* \in \mathbb{R}^{d_1 \times 1}$ are randomly initialized with small values
**10** **end**
**11** **else**
**12**     $\boldsymbol{A}_i^t \leftarrow \boldsymbol{A}_i^t, \boldsymbol{B}_i^t \leftarrow \boldsymbol{B}_i^t$
**13** **end**

---

■ **Rank Bonus Chance via Orthogonal Subspace Projection.** CP-Rank leverages the task subspace similarity metric to distinguish the low-rank subspaces of the new task $\mathcal{T}_i$ into two categories: (i) in the orthogonal region $OR_i$, where the subspaces are orthogonal to the previous tasks $\{\mathcal{T}_m\}_{m=1}^{i-1}$, (ii) in the non-orthogonal region $OR_i^{\perp}$, where the subspaces are not orthogonal to the prior tasks $\{\mathcal{T}_m\}_{m=1}^{i-1}$. For the low-rank layerwise matrices $\boldsymbol{B}_i^l \boldsymbol{A}_i^l$ in $OR_i$, CP-Rank safely increases the rank of them to enhance generalization. However, for those low-rank layerwise matrices $\boldsymbol{B}_i^l \boldsymbol{A}_i^l$ in $OR_i^{\perp}$, CP-Rank halts rank growth of them, as these subspaces may interfere with the subspaces of previously learned tasks in an intriguing manner. Thus, to reduce the interference of the low-rank subspaces in $OR_i^{\perp}$, we apply orthogonal gradient projection for the low-rank matrix update instead of SGD update. By progressively using orthogonal updates, more low-rank subspaces would shift in $OR_i$ for task $\mathcal{T}_i$, allowing them to obtain the bonus chance to increase their ranks and thus improve generalization. We utilize the low-rank structure of LoRA parameters, which suggests that they encapsulate critical update directions rather than merely acting as numerical adjustments Wang et al. (2023), meaning that the gradient subspaces of previous tasks are effectively captured by LoRA parameters, thus reducing computation and memory. Instead of directly ensuring the orthogonality of $\boldsymbol{A}_i^l$ as in Wang et al. (2023), we consider $\boldsymbol{B}_i^l \boldsymbol{A}_i^l$ due to additional random parameters for $\boldsymbol{B}_i^l$ during training and enforce the orthogonality through the left singular matrices of task $\mathcal{T}_i$ and previous tasks $\{\mathcal{T}_m\}_{m=1}^{i-1}$ during the training of task $\mathcal{T}_i$:

$$\sum_{(\boldsymbol{x},y) \in \mathcal{T}_i} \log p_{\boldsymbol{\theta}}(y|\boldsymbol{x}) + \lambda_1 \sum_{l=1}^{L} \sum_{m=1}^{i-1} \sum_{j,k} \|[(\boldsymbol{U}_i^l)^{\top} \boldsymbol{U}_m^l]_{j,k}\|^2 \tag{8}$$

where $[\boldsymbol{U}_i^l, \boldsymbol{U}_m^l]_{j,k}$ denotes the element at $j$-th row and $k$-th column of $(\boldsymbol{U}_i^l)^{\top} \boldsymbol{U}_m^l$. Here we use top $r_i^l$ singular vectors of $\boldsymbol{U}_i^l$ and top $r_m^l$ singular vectors of $\boldsymbol{U}_m^l$ to achieve the orthogonality.

## 3 EXPERIMENTS

### 3.1 EXPERIMENTAL SETUP

Our experiments utilize the encoder-decoder architecture of the T5-large and T5-base models Raffel et al. (2020), in line with previous work in continual learning (CL) for NLP. All experiments are conducted on NVIDIA A6000 GPUs, leveraging the DeepSpeed repository.

#### 3.1.1 DATASETS

■ **Standard CL benchmark.** We evaluate our approach on a standard CL benchmark designed specifically for language models, comprising five text classification datasets: AG News, Amazon Reviews, Yelp Reviews, DBpedia, and Yahoo Answers, as introduced by Zhang et al. (2015). We follow the CL setup for the T5 model outlined in LFPT5 Qin & Joty (2021), experimenting with three different task orders within this benchmark.

■ **Large number of tasks.** To further assess the effectiveness of our method, we evaluate it on extended task sequences using a comprehensive CL benchmark that involves 15 datasets, as described in Razdaibiedina et al. (2023). This benchmark combines tasks from three distinct sources: five from the standard CL benchmark, four from the GLUE benchmark (MNLI, QQP, RTE, SST-2), five from the SuperGLUE benchmark (WiC, CB, COPA, MultiRC, BoolQ), and the IMDB movie reviews dataset. For each task, we train on 1000 randomly selected samples and validate using 500 samples per class, adhering to the methodology of Razdaibiedina et al. (2023).

■ **Math benchmarks.** We test the performance of our method on challenging math benchmarks, specifically GSM8K Cobbe et al. (2021) and MATH Hendrycks et al. (2021). GSM8K includes a collection of $8.5K$ graduate-school math word problems and the solutions of these problems perform a sequence of elementary calculations using basic arithmetic operations and natural language. MATH consists of problems from mathematics competitions, covering a range of difficulty levels in areas such as Algebra, Counting & Probability, Geometry, Intermediate Algebra, Number Theory, Prealgebra, and Precalculus, with solutions written in LaTeX and natural language. For both GSM8K and MATH benchmarks, we train on 7500 examples, testing GSM8K on 1000 examples and MATH on 5000 examples.

### 3.1.2 METRICS

We define the testing accuracy on task $\mathcal{T}_i$ after training on task $\mathcal{T}_j$ as $a_{i,j}$. The primary evaluation metric is **Average Accuracy (AA)**, which is computed as the mean accuracy across all tasks after completing the training on the final task: $\frac{1}{T} \sum_{i=1}^{T} a_{i,T}$.

### 3.1.3 BASELINES

We compare our method against various baseline approaches:

- SeqFT de Masson D'Autume et al. (2019): train all model parameters on a sequence of tasks (without adding any regularization or replaying samples from the previous tasks).
- SeqLoRA: fixed-size LoRA parameters are trained on a sequence of tasks (without adding any regularization or replaying samples from the previous tasks).
- IncLoRA: incremental learning of new LoRA parameters on a sequence of tasks (without adding any regularization or replaying samples from the previous tasks).
- Replay: fine-tune the whole model with a memory buffer, and replay samples from old tasks when learning new tasks to avoid forgetting.
- EWC Kirkpatrick et al. (2017): fine-tune the whole model with a regularization loss that prevents updating parameters that could interfere with previously learned tasks.
- LwF Li & Hoiem (2017): constrains the shared representation layer to be similar to its original state before learning the new task.
- L2P Wang et al. (2022): uses the input to dynamically select and update prompts from the prompt pool in an instance-wise fashion.
- LPT5 Qin & Joty (2021): continuously train a soft prompt that simultaneously learns to solve the tasks and generate training samples, which are subsequently used in experience replay.
- ProgPrompt Razdaibiedina et al. (2023): adopts task-specific soft prompts for each task, training distinct models per task and using task IDs during inference.
- O-LoRA Wang et al. (2023): incrementally train new tasks in an orthogonal subspace while fixing the LoRA matrices of previous tasks.
- PerTaskFT: train a separate model for each task individually.
- MTL: train a multi-task learning model on all tasks simultaneously, serving as the performance upper bound for the benchmark.

### 3.2 MAIN RESULTS

Tab. 1 presents the performance comparisons of CP-Rank and baseline continual learning methods across two CL benchmarks. In line with LFPT5, we report the average results from three random runs, each with a different task order on the CL benchmark.

■ **Results on Standard Continual Learning Benchmarks.** Across three task orders of the standard CL benchmark, CP-Rank with orthogonal projection ('CP-Rank w OP' in Tab. 1) consistently outperforms previous methods by a significant margin. Specifically, CP-Rank with orthogonal pro-

jection shows performance improvements across all task orders compared to O-LoRA, the prior state-of-the-art. Furthermore, CP-Rank without orthogonal projection ('CP-Rank wo OP' in Tab. 1) exceeds other previous methods except O-LoRA. Our approach also achieves performance comparable to multi-task learning (MTL) and surpasses PerTaskFT by a notable margin. This demonstrates that CP-Rank with orthogonal projection not only effectively mitigates catastrophic forgetting but also efficiently leverages prior task knowledge to enhance the learning of new tasks.

■ **Results on Large Number of Tasks.** In a more demanding benchmark featuring a large number of tasks, CP-Rank with orthogonal projection surpasses the state-of-the-art, O-LoRA, in terms of average performance across the three task orders. Notably, CP-Rank without orthogonal projection also exceeds IncLoRA, as our method relies solely on increasing the training rank based on the subspace similarity metric and updating interval, compared to IncLoRA. While ProgPrompt shows strong performance in long task sequences, it has significant limitations. ProgPrompt is strictly tied to the tasks it is trained on and depends heavily on task IDs during inference, which limits its generalization and adaptability for LLMs. In contrast, our method does not require task IDs during testing, making it more generalizable. However, it is worth noting that nearly all existing continual learning methods still fall considerably short of the performance levels achieved by PerTaskFT and MTL, underscoring the challenges of continual learning with a large number of tasks.

Table 1: Comparison of testing performance on two standard CL benchmarks using the T5-large model across different task orders. We report the average testing accuracy after training the final task in each task order, averaged over three random runs.

| Order | Standard CL Benchmark | | | | Large Number of Tasks | | | |
|---|---|---|---|---|---|---|---|---|
| | 1 | 2 | 3 | avg | 4 | 5 | 6 | avg |
| SeqFT | 18.9 | 24.9 | 41.7 | 28.5 | 7.4 | 7.3 | 7.4 | 7.4 |
| SeqLoRA | 39.5 | 31.9 | 46.6 | 39.3 | 4.9 | 3.5 | 4.2 | 4.2 |
| IncLoRA | 63.4 | 62.2 | 65.1 | 63.6 | 63.0 | 57.9 | 60.4 | 60.5 |
| Replay | 50.3 | 52.0 | 56.6 | 53.0 | 54.5 | 54.3 | 53.5 | 54.1 |
| EWC | 46.3 | 45.3 | 52.1 | 47.9 | 44.9 | 44.0 | 45.4 | 44.8 |
| LwF | 52.7 | 52.9 | 48.4 | 51.3 | 49.7 | 42.8 | 46.9 | 46.5 |
| L2P | 59.0 | 60.5 | 59.9 | 59.8 | 57.7 | 53.6 | 56.6 | 56.0 |
| LFPT5 | 66.6 | 71.2 | 76.2 | 71.3 | 69.8 | 67.2 | 69.2 | 68.7 |
| CP-Rank(wo OP) | 72.8 | 73.7 | 70.7 | 72.4 | 63.2 | 65.2 | 62.1 | 63.5 |
| O-LoRA | 74.9 | 75.3 | 75.9 | 75.4 | **70.5** | 65.5 | 70.5 | 68.8 |
| CP-Rank(w OP) | **77.3** | **77.1** | **76.0** | **76.8** | 69.9 | **69.2** | **71.5** | **70.2** |
| ProgPrompt | 76.1 | 76.0 | 76.3 | 76.1 | 78.7 | 78.8 | 77.8 | 78.4 |
| PerTaskFT | 70.0 | 70.0 | 70.0 | 70.0 | 78.1 | 78.1 | 78.1 | 78.1 |
| MTL | 80.0 | 80.0 | 80.0 | 80.0 | 76.3 | 76.3 | 76.3 | 76.3 |

## 3.3 IMPACT ON THE FORGETTING PERFORMANCE ON MATH DATASETS

We investigate our method on two challenging math datasets, GSM8K and MATH. Each dataset is evenly split into two subsets, creating a total of four tasks. We experiment with two distinct task orders to assess the impact on forgetting: one where the model first trains on GSM8K followed by MATH, and another where the task order is reversed, starting with MATH and then proceeding to GSM8K. This setup allows us to evaluate how task sequence influences the model's forgetting performance, and our average testing accuracy is the exact prediction accuracy of the final answer for each task, aligned with other works (Chung et al., 2024; Magister et al., 2023). Tab. 2 presents the testing accuracy trends for each dataset after training on successive tasks. Across both task orders, CP-Rank with orthogonal projection consistently outperforms O-LoRA in terms of final average testing accuracy. Moreover, CP-Rank demonstrates less forgetting on the first task and better generalization on the second task compared to O-LoRA. This suggests that CP-Rank is more effective at mitigating forgetting, especially when switching between tasks in these math benchmarks. Additionally, we need to note that fully fine-tuning MATH using the T5-large model achieves approximately $3.0\%$, while fully fine-tuning GSM8K using the T5-large model approaches $4.2\%$, as also reported in the work Magister et al. (2023).

Table 2: Comparison of testing accuracy changes on GSM8K and MATH datasets when using CP-Rank and O-LoRA, with the T5-large model trained on sequential tasks from GSM8K and MATH.

| | $\mathcal{T}_1$: **GSM8K** $\rightarrow$ $\mathcal{T}_2$: **MATH** $\rightarrow$ $\mathcal{T}_3$: **GSM8K** $\rightarrow$ $\mathcal{T}_4$: **MATH** | | | | |
| **Method** | **GSM8K** | **MATH** | **avg** | PerTaskFT | MTL |
|---|---|---|---|---|---|
| O-LoRA | $1.87 \rightarrow 0.01 \rightarrow 0.22 \rightarrow 0.23$ | $1.18 \rightarrow 1.34 \rightarrow 1.6$ | 1.32 | 3.63 | 3.88 |
| CP-Rank | $1.87 \rightarrow 0.08 \rightarrow 0.76 \rightarrow 1.1$ | $1.58 \rightarrow 2.32 \rightarrow 2.38$ | **2.23** | 3.63 | 3.88 |
| | $\mathcal{T}_1$: **MATH** $\rightarrow$ $\mathcal{T}_2$: **GSM8K** $\rightarrow$ $\mathcal{T}_3$: **MATH** $\rightarrow$ $\mathcal{T}_4$: **GSM8K** | | | | |
| **Method** | **MATH** | **GSM8K** | **avg** | PerTaskFT | MTL |
| O-LoRA | $2.62 \rightarrow 1.44 \rightarrow 2.34 \rightarrow 2.32$ | $0.30 \rightarrow 0.15 \rightarrow 0.14$ | 1.87 | 3.63 | 3.88 |
| CP-Rank | $2.62 \rightarrow 2.54 \rightarrow 2.62 \rightarrow 2.48$ | $1.21 \rightarrow 1.14 \rightarrow 0.38$ | **2.04** | 3.63 | 3.88 |

## 3.4 DISCUSSIONS

■ **What's the resulting rank distribution across different layers?** Fig. 2a and Fig. 2b show the sum of the resulting rank of low-rank matrices trained on the last three tasks of Order 3 in the standard CL benchmark since the rank for the first task in our setting is not affected by other tasks. We find that the rank distribution in the $v$ modules varies more than in the $q$ modules of encoder layers, while in the decoder layers, the $q$ and $v$ modules exhibit different patterns of variability. Meanwhile, the rank distribution in the encoder layers is slightly more consistent compared to the decoder layers. These findings suggest that different layers within the model fulfill distinct roles, which our method effectively leverages to achieve better overall performance.

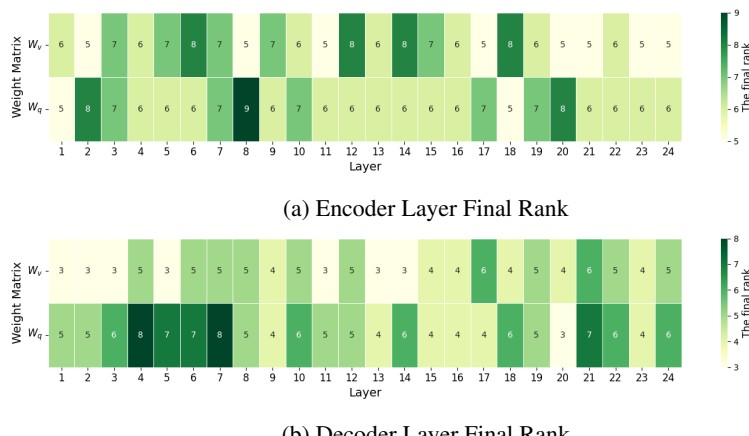

(a) Encoder Layer Final Rank

(b) Decoder Layer Final Rank

Figure 2: Comparison of the sum of the resulting ranks in different modules of encoder and decoder layers in CP-Rank training last three tasks of order 3 on standard CL benchmark. Here the $x$-axis is the layer index and the $y$-axis represents different modules (types) of low-rank weight matrices.

■ **How do different fixed ranks perform in different settings?** Tab. 3 presents the average testing accuracy in different fixed rank settings compared with CP-Rank. CP-Rank with orthogonal projection, where we use rank 8 for the first task same as the setting in O-LoRA, consistently outperforms these fixed rank settings. In both IncLoRA and O-LoRA settings, increasing the rank improves the average accuracy of the model to a certain extent. Specifically, in O-LoRA, there is not a significant difference in performance between $r = 2$ and $r = 6$, while in IncLoRA, there is an evident gap between $r = 2$ and $r = 8$ but the difference between $r = 8$ and $r = 16$ is not significant. This suggests that in IncLoRA, $r = 2$ is insufficient for effective learning and generalization without any aid of CL techniques. Moreover, when comparing the rank usage across different settings, CP-Rank achieves better results with an average rank usage that falls between $r = 2$ and $r = 4$, making it more memory-efficient while still delivering superior performance.

Table 3: Comparison of different rank patterns across methods using T5-large model on the standard CL benchmark. The average rank refers to the sum of incremental LoRA ranks across all layers for the last three tasks, with CP-Rank computing the sum of ranks across all layers to obtain the average.

| | Order | | | | |
|---|---|---|---|---|---|
| | **1** | **2** | **3** | **avg** | **r(avg)/$\mathcal{T}$** |
| **CP-Rank w OP** | **77.3** | **77.1** | **76.0** | **76.8** | 3 |
| **CP-Rank wo OP** | 72.8 | 73.7 | 70.7 | 72.4 | 4 |
| **IncLoRA** ($r=2$) | 44.5 | 48.5 | 50.7 | 47.9 | 2 |
| **IncLoRA** ($r=4$) | 50.4 | 44.0 | 56.7 | 50.4 | 4 |
| **IncLoRA** ($r=8$) | 63.4 | 62.2 | 65.1 | 63.6 | 8 |
| **IncLoRA** ($r=16$) | 62.5 | 62.4 | 67.5 | 64.1 | 16 |
| **O-LoRA** ($r=2$) | 73.5 | 73.2 | 74.4 | 73.7 | 2 |
| **O-LoRA** ($r=4$) | 75.7 | 75.6 | 75.4 | 75.6 | 4 |
| **O-LoRA** ($r=8$) | 74.9 | 75.3 | 75.9 | 75.4 | 8 |
| **O-LoRA** ($r=16$) | 75.2 | 74.9 | 76.9 | 75.7 | 16 |

Table 4: Comparison of CP-Rank performance across three task orders of the standard CL benchmark with varying fixed ranks for the first task, using T5-large model.

| | Order | | | |
|---|---|---|---|---|
| **Rank** | **1** | **2** | **3** | **avg** |
| $r_1 = 2$ | 76.7 | 76.3 | 76.1 | 76.4 |
| $r_1 = 4$ | 76.9 | 76.4 | 76.1 | 76.5 |
| $r_1 = 8$ | 77.3 | 77.1 | 76.0 | 76.8 |
| $r_1 = 16$ | 77.2 | 76.7 | 76.3 | 76.7 |

■ **How does the fixed rank of the first task affect CP-Rank?** Tab. 4 shows the performance of CP-Rank with different fixed ranks of the first task. We use CP-Rank with orthogonal projection to evaluate the standard CL benchmark. It suggests that increasing the fixed rank for the first task might slightly improve the final average accuracy but differences between different ranks are relatively modest, where there is not a significant gap between $r_1 = 2$ and $r_1 = 16$. It indicates that CP-Rank does not heavily depend on the specific rank of the first task but maintain robust performance across a variety of initial rank settings.

■ **How does subspace similarity threshold $\epsilon$ affect the performance of CP-Rank?** We evaluate the performance for different values of $\epsilon$ (0.001, 0.005, 0.01, 0.05, 0.1, 0.5) as shown in Tab. 5. The results indicate that accuracy remains very stable at $\epsilon = 0.001$ for CP-Rank with orthogonal projection, and at $\epsilon = 0.1$ for CP-Rank without orthogonal projection. There is a slight downtrend between 0.001 and 0.1 in CP-Rank with orthogonal projection, since with orthogonal projection, more subspaces are orthogonal to the previously learned subspaces and a larger threshold would make negative-affected subspaces increase in rank thus worsen the results. In CP-Rank without orthogonal projection, the accuracy at 0.5 is slightly better than 0.1, suggesting that more subspaces are treated as orthogonal to generalize better via increasing rank.

Table 5: Comparison of CP-Rank Performance on different $\epsilon$ values across three task orders in standard CL benchmark.

| | Impact of threshold $\epsilon$ | | | | | | | |
|---|---|---|---|---|---|---|---|---|
| **CP-Rank w OP** | 0.001 | 0.005 | 0.01 | 0.05 | 0.1 | **CP-Rank wo OP** | 0.1 | 0.5 |
| **Order 1** | 77.3 | 78.3 | 75.0 | 76.8 | 75.7 | **Order 1** | 70.0 | 72.8 |
| **Order 2** | 77.1 | 77.3 | 76.3 | 75.4 | 76.7 | **Order 2** | 70.8 | 73.7 |
| **Order 3** | 77.1 | 76.7 | 76.2 | 74.3 | 73.6 | **Order 3** | 69.9 | 70.7 |

■ **How does updating intervals $k$ work for CP-Rank?** Hyper-parameter $k$ controls the frequency of subspace similarity threshold $\epsilon$. To analyze the effect of $k$, we vary $k$ in $50, 60, 70, 80, 90$ by keeping other hyper-parameters the same. Fig. 3 shows that the accuracy at different $k$ is stable in CP-Rank with orthogonal projection, while the performance of CP-Rank without orthogonal projection performs a little stable from 50 to 80 but drops sharply at 90, since in this case, larger updating intervals cannot grasp the rapid changes in subspaces and would miss critical changing points.

■ **How do different pre-trained models influence performances?** We investigate the impact of model scale on performance by comparing T5-base and T5-large models on standard CL benchmark. We evaluate both CP-Rank with orthogonal projection and O-LoRA across three task orders. The results, shown in Tab. 6, present the performance differences between the two model sizes and the methods employed. For T5-base model, CP-Rank with orthogonal projection consistently outperforms O-LoRA. While for T5-large model, CP-Rank significantly surpasses O-LoRA's outcomes. Moreover, CP-Rank shows exceptional consistency across all task orders in T5-large model, highlighting its robustness and effectiveness when the model size is scaled up.

Figure 3: Comparison of different updating interval performances across three task orders in the standard CL benchmark.

Table 6: Comparison of different models' performances across three task orders in standard CL benchmark.

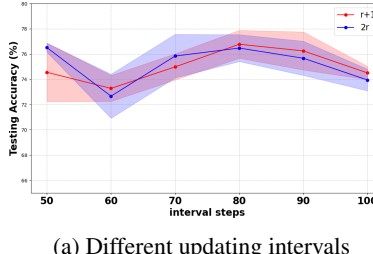

| T5-base | Order | | | | |
|---|---|---|---|---|---|
| | 1 | 2 | 3 | avg | MTL |
| O-LoRA | 72.9 | 72.3 | **72.6** | 72.6 | 78.3 |
| CP-Rank | **74.0** | **72.7** | 72.1 | **72.9** | 78.3 |
| T5-large | 1 | 2 | 3 | avg | MTL |
| O-LoRA | 74.9 | 75.3 | 75.9 | 75.4 | 80.0 |
| CP-Rank | **77.3** | **77.1** | **76.0** | **76.8** | 80.0 |

■ **What's the difference between different increasing rules ($2r$ v.s. $r + 1$)?** To evaluate effectiveness of our updating rule $\text{update}(r) = r + 1$, we compare it with another common updating rule $\text{update}(r) = 2r$, as mentioned in the work Cosson et al. (2022), in some cases, $r + 1$ can be advantageously replaced by $2r$. Fig. 4 shows the performance of two different updating rules with different updating intervals $k$ and subspace similarity threshold $\epsilon$. The changing patterns of two updating rules are almost overlapped across different updating intervals. In different subspace similarity thresholds, in the range from $0.001$ to $0.01$, updating rule $r + 1$ is less variable while $2r$ drops sharply at $0.005$. As the threshold increases, updating rule $r + 1$ experiences a performance drop and becomes variable, but $2r$ remains stable at lower accuracy. These results suggest that $r + 1$ performs better within a certain threshold range, which is why we chose it as our updating rule.

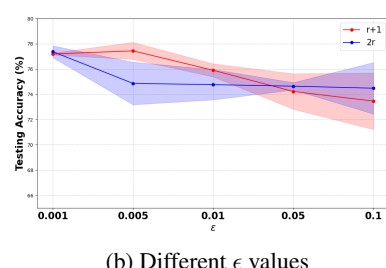

(a) Different updating intervals      (b) Different $\epsilon$ values

Figure 4: Comparison of different increasing rules across three task orders in standard CL benchmark using T5-large model in terms of updating intervals and subspace similarity thresholds.

## 4 CONCLUSION

We propose a parameter-efficient continual learning method, CP-Rank, which gradually increases the layerwise rank of incremental LoRAs for new tasks based on the between-task low-rank subspace similarity metric. CP-Rank not only accounts for the low-rank relationships between tasks' incremental LoRAs but also adapts to the unique low-rank dynamics across different model layers. This approach effectively mitigates forgetting of previous tasks while enhancing generalization on new tasks in a memory-efficient way. We perform extensive experiments on both natural language processing and challenging math reasoning tasks, demonstrating that CP-Rank captures rank patterns effectively in CL and consistently outperforms existing methods.

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

# A APPENDIX

## A.1 ADDITIONAL RELATED WORKS

**Continual Learning.** Continual learning aims to develop algorithms that can continuously accumulate and refine knowledge, especially when handling dynamic data streams. The key challenge is overcoming catastrophic forgetting, where a model's performance on previously learned tasks significantly declines after being trained on new tasks. To tackle this issue, existing approaches are generally classified into three main categories: (i) *Rehearsal-based methods*, which use a memory buffer to retain data samples from previous tasks, incorporating techniques such as experience replay Rolnick et al. (2019), or constrained optimization to allow the model to learn from current and previous tasks simultaneously (Lopez-Paz & Ranzato, 2017; Han et al., 2020). (ii) *Regularization-based methods*, which add extra terms to the loss function to penalize changes in important model parameters, limiting interference with previously learned tasks (Kirkpatrick et al., 2017; Li & Hoiem, 2017; Farajtabar et al., 2020; Smith et al., 2023). For example, EWC Kirkpatrick et al. (2017) preserves knowledge of old tasks by slowing down learning on weights deemed important for those tasks, while OGD Farajtabar et al. (2020) ensures that parameters move within the orthogonal space defined by previous task gradients. (iii) *Architecture-based methods*, which aim to reduce task interference by dynamically expanding the model's capacity or creating separate components for each task (Rusu et al., 2016; Yoon et al., 2017; Li et al., 2019; Rao et al., 2019; Razdaibiedina et al., 2023). For instance, Progressive Prompts Razdaibiedina et al. (2023) improves forward transfer and mitigates forgetting by learning a distinct prompt for each new task and sequentially appending these task-specific prompts to previously learned ones.

**Parameter-efficient Tuning.** Recent works on parameter efficient tuning (PET) He et al. (2021) have demonstrated that training only a subset of model parameters can achieve performance comparable to full model fine-tuning, while significantly reducing computational and annotation costs (Zaken et al., 2021; Lester et al., 2021; Houlsby et al., 2019; Hu et al., 2021; Zhang et al., 2023b). For instance, BitFit Zaken et al. (2021) finds that shows that updating only the bias terms during fine-tuning is highly effective. Prompt tuning Lester et al. (2021) leverages learnable 'soft prompts' via back-propagation to condition frozen language models for specific tasks. LoRA Hu et al. (2021) employs low-rank adapters to adapt models to new tasks with minimal additional parameters, and AdaLoRA Zhang et al. (2023b) builds on LoRA by dynamically allocating the parameter budget based on the importance of the weight matrices. While most PET methods focus on learning a single task, some efforts have extended PET to continual learning. AdapterCLMadotto et al. (2020)introduces a dedicated adapter block for each task, and LFPT5 Qin & Joty (2021) continuously trains a large soft prompt across multiple tasks. ConPET Song et al. (2023) adapts existing continual learning strategies—originally developed for smaller models—to LLMs by integrating PET with a dynamic replay mechanism. O-LoRA Wang et al. (2023) incrementally learns new tasks in orthogonal subspaces, keeping LoRA parameters from previous tasks fixed to mitigate catastrophic forgetting. However, O-LoRA uses the fixed same rank for all incremental LoRAs without investigating the rank patterns in CL.

## A.2 IMPLEMENTATION DETAILS

All experiments with T5 models were conducted on a server equipped with four NVIDIA A6000 GPUs, using the DeepSpeed library for efficient implementation. Across all task sequences and different task orders, we maintained a consistent experimental setup: the learning rate was set to 1e-3, with a total batch size of 32, distributed as 8 per GPU to fully utilize the computational power of the A6000 GPUs. We applied a dropout rate of 0.1, while no additional weight penalty (0.0 weight decay) was imposed during training.

## A.3 DATASETS

### A.3.1 CONTINUAL LEARNING BENCHMARKS

Tab. 7 provides detailed information on the 15 datasets used in our continual learning (CL) experiments, along with the evaluation metrics employed. The selected datasets include those from well-

established benchmarks: the standard CL benchmark Zhang et al. (2015), GLUE Wang et al. (2018), and SuperGLUE benchmarks Wang et al. (2019), as well as the IMDB movie reviews dataset.

| Dataset name | Category | Task | Domain | Metric |
|---|---|---|---|---|
| 1. Yelp | CL Benchmark | Sentiment Analysis | Yelp Reviews | Accuracy |
| 2. Amazon | CL Benchmark | Sentiment Analysis | Amazon Reviews | Accuracy |
| 3. DBpedia | CL Benchmark | Topic Classification | Wikipedia | Accuracy |
| 4. Yahoo | CL Benchmark | Topic Classification | Yahoo Q&A | Accuracy |
| 5. AG News | CL Benchmark | Topic Classification | News | Accuracy |
| 6. MNLI | GLUE | NLI | Various | Accuracy |
| 7. QQP | GLUE | Paragraph Detection | QUora | Accuracy |
| 8. RTE | GLUE | NLI | News, Wikipedia | Accuracy |
| 9. SST-2 | GLUE | Sentiment Analysis | Movie Reviews | Accuracy |
| 10. WiC | SuperGLUE | Word Sense Disambiguation | Lexical Databases | Accuracy |
| 11. CB | SuperGLUE | NLI | Various | Accuracy |
| 12. COPA | SuperGLUE | QA | Blogs,Encyclopedia | Accuracy |
| 13. BoolQA | SuperGLUE | Boolean QA | Wikipedia | Accuracy |
| 14. MultiRC | SuperGLUE | QA | Various | Accuracy |
| 15. IMDB | SuperGLUE | Sentiment Analysis | Movie Reviews | Accuracy |

Table 7: The details of 15 datasets used in our CL experiments. NLI denotes natural language inference, QA denotes questions and answers task. The first five tasks correspond to the standard CL benchmark, all other tasks are used in long-sequence experiments

| Order | Model | Task Sequence |
|---|---|---|
| 1 | T5-large,T5-base | dbpedia$\to$ amazon $\to$ yahoo $\to$ ag |
| 2 | T5-large,T5-base | dbpedia$\to$ amazon $\to$ ag$\to$ yahoo |
| 3 | T5-large,T5-base | yahoo $\to$ amazon $\to$ ag $\to$ dbpedia |
| 4 | T5-large | mnli $\to$ cb $\to$ wic $\to$ copa $\to$ qqp $\to$ boolqa $\to$ rte $\to$ imdb $\to$ yelp $\to$ amazon $\to$ sst-2 $\to$ dbpedia $\to$ ag $\to$ multirc $\to$ yahoo |
| 5 | T5-large | multirc $\to$ boolqa $\to$ wic $\to$ mnli $\to$ cb $\to$ copa $\to$ qqp $\to$ rte $\to$ imdb $\to$ sst-2 $\to$ dbpedia $\to$ ag $\to$ yelp $\to$ amazon $\to$ yahoo |
| 6 | T5-large | yelp $\to$ amazon $\to$ mnli $\to$ cb $\to$ copa $\to$ qqp $\to$ rte $\to$ imdb$\to$ sst-2 $\to$ dbpedia $\to$ ag $\to$ yahoo $\to$ multirc $\to$ boolqa $\to$ wic |

Table 8: Six different task sequence orders utilized in continual learning experiments. Orders 1-3 follow the standard continual learning benchmark as established by previous research, focusing on a more traditional task sequence. Orders 4-6 customized for long-sequence experimentation, encompass 15 tasks each and are structured according to the methodologies outlined in Razdaibiedina et al. (2023).

### A.3.2 MATH BENCHMARKS

Tab. 10 and Tab. 11 shows the data structure of both GSM8K and MATH, and Tab. 12 provides the task information and evaluation metric for math datasets.

### A.4 RESULTING RANK DISTRIBUTIONS ACROSS DIFFERENT LAYERS

Fig. 5 and Fig. 6 show the sum of the final learned ranks of the last three tasks in order 1 and order 2 for the standard CL benchmark. We find that the rank distribution in the $v$ modules varies more than in the $q$ modules of encoder layers, while in the decoder layers, the $q$ and $v$ modules exhibit different patterns of variability. Meanwhile, the rank distribution in the encoder layers is slightly more consistent compared to the decoder layers. These findings suggest that different layers within

| Task | Prompts |
|------|---------|
| NLI | What is the logical relationship between the "sentence 1" and the "sentence 2"? Choose one from the option. |
| QQP | Whether the "first sentence" and the "second sentence" have the same meaning? Choose one from the option. |
| SC | What is the sentiment of the following paragraph? Choose one from the option. |
| TC | What is the topic of the following paragraph? Choose one from the option. |
| BoolQA | According to the following passage, is the question true or false? Choose one from the option. |
| MultiRC | According to the following passage, is the question true or false? Choose one from the option. |
| WiC | Given a word and two sentences, whether the word is used with the same sense in both sentences? Choose one from the option. |

Table 9: Instructions for different tasks

| Data Field | Data Content |
|------------|--------------|
| **question** | Natalia sold clips to 48 of her friends in April, and then she sold half as many clips in May. How many clips did Natalia sell altogether in April and May? |
| **answer** | Natalia sold $48/2 = << 48/2 = 24 >> 24$ clips in May. Natalia sold $48 + 24 = << 48 + 24 = 72 >> 72$ clips altogether in April and May. $\#\#\#\#72$ |

Table 10: Data structure of GSM8K dataset

the model fulfill distinct roles, which our method effectively leverages to achieve better overall performance.

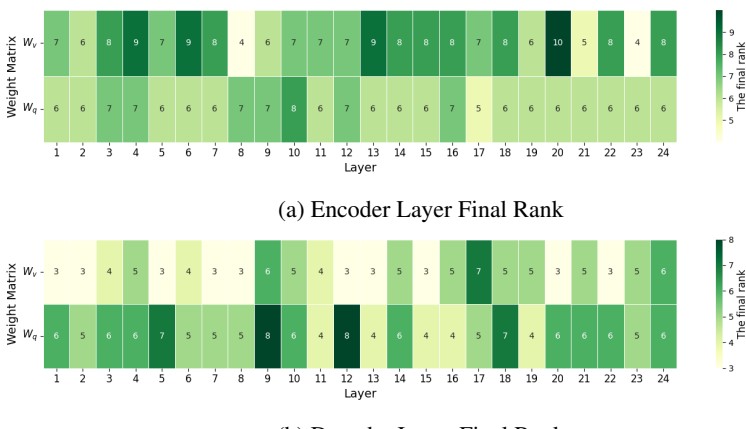

(a) Encoder Layer Final Rank

(b) Decoder Layer Final Rank

Figure 5: Comparison of the sum of the resulting ranks in the different modules of encoder and decoder Layers in CP-Rank training last three tasks of order 1 on standard CL benchmark. Here the $x$-axis is the layer index and the $y$-axis represents different modules (types) of low-rank weight matrices.

| Data Field | Data Content |
|---|---|
| **problem** | A board game spinner is divided into three parts labeled $A$, $B$ and $C$. The probability of the spinner landing on $A$ is $\frac{1}{3}$ and the probability of the spinner landing on $B$ is $\frac{5}{12}$. What is the probability of the spinner landing on $C$? Express your answer as a common fraction. |
| **level** | Level 1 |
| **type** | Counting & Probability |
| **solution** | The spinner is guaranteed to land on exactly one of the three regions, so we know that the sum of the probabilities of it landing in each region will be 1. If we let the probability of it landing in region $C$ be $x$, we then have the equation $1 = \frac{5}{12}+\frac{1}{3}+x$, from which we have $x=\boxed{\frac{1}{4}}$. |

Table 11: Data structure of MATH dataset

| Dataset name | Category | Task | Metric |
|---|---|---|---|
| 1. GSM8K | Math Benchmark | Math Reasoning | Exact Prediction Accuracy |
| 2. MATH | Math Benchmark | Math Reasoning | Exact Prediction Accuracy |

Table 12: The details of GSM8K and MATH datasets used in our CL experiments. For the matric, we use the exact prediction accuracy for evaluating these two datasets, which is the correction rate of the final answer.

## A.5 Forgetting Error in low-rank CL

In this section, we examine the forgetting error is a toy setting to illustrate the proposed method. This analysis aims to elucidate the key idea to gradually increase the rank to mitigate forgetting. It is important to note that this analysis is not intended to be rigorous and future work will focus on developing a more thorough and rigorous understanding of the forgetting and generalization errors. To this end, inspired by Li et al. (2023), we consider a simple linear regression setting with two tasks. Each task $\mathcal{T}_k$ is associated with a data distribution $\mathcal{D}_k$ and contains a separate target dataset $\mathcal{S}_k = \{(\boldsymbol{x}_{k,i}, y_{k,i})\}_{i=1}^{n_k}$ where $\boldsymbol{x}_{k,i} \in \mathcal{X}_k$ and $y_{k,i} \in \mathcal{Y}_k$. For simplicity, we use 2 tasks and all $n_k = n$ for explanation. The population risks for the two tasks can be denoted by

$$\mathcal{R}_1(\boldsymbol{W}_0 + \boldsymbol{B}\boldsymbol{A}) = \frac{1}{n}\mathbb{E}_{\mathcal{D}_1}\|y_1 - \boldsymbol{X}_1(\boldsymbol{W}_0 + \boldsymbol{B}\boldsymbol{A})\|^2 \tag{9}$$

$$\mathcal{R}_2(\boldsymbol{W}_0 + \boldsymbol{B}\boldsymbol{A}) = \frac{1}{n}\mathbb{E}_{\mathcal{D}_2}\|y_2 - \boldsymbol{X}_2(\boldsymbol{W}_0 + \boldsymbol{B}\boldsymbol{A})\|^2 \tag{10}$$

respectively, where $(\boldsymbol{X}_1, y_1)$ is the dataset of the first task and $(\boldsymbol{X}_2, y_2)$ is the dataset of the second task. $\boldsymbol{X}_k = (\boldsymbol{x}_{k,1}, \ldots, \boldsymbol{x}_{k,n}) \in \mathbb{R}^{n \times d}$ and $y_K = (y_{k,1}, \ldots, y_{k,n}) \in \mathbb{R}^n$ for $k = 1, 2$.

**Assumption 1** (Fixed design) Assume that the feature vectors $(\boldsymbol{x}_{1,i})_{i=1}^n$ and $(\boldsymbol{x}_{2,i})_{i=1}^n$ are fixed and that the labels $(y_{1,i})_{i=1}^n$ and $(y_{2,i})_{i=1}^n$ are independent random variables.

**Assumption 2** (Shared optimal parameter) Assume that there exists a $\boldsymbol{B}^*\boldsymbol{A}^*$ where $\boldsymbol{B}^* \in \mathbb{R}^{d_1 \times r}$, $\boldsymbol{A}^* \in \mathbb{R}^{r \times d_2}$ such that

$$\boldsymbol{B}^*\boldsymbol{A}^* \in \arg\min \mathcal{R}_1(\boldsymbol{W}_0 + \boldsymbol{B}\boldsymbol{A}), \quad \boldsymbol{B}^*\boldsymbol{A}^* \in \arg\min \mathcal{R}_2(\boldsymbol{W}_0 + \boldsymbol{B}\boldsymbol{A}) \tag{11}$$

We assume there is a common optimal low-rank parameter for the two tasks, which follows (Li et al., 2023; Van de Ven & Tolias, 2019; Evron et al., 2022).

**Assumption 3** (Well-specified noise) Assume that for $\boldsymbol{B}^*\boldsymbol{A}^*$ in Assumption 2, it holds that: for $k = 1, 2$ and $i = 1, \ldots, n$,

$$\mathbb{E}[y_{k,i}] = \boldsymbol{x}_{k,i}^\top(\boldsymbol{W}_0 + \boldsymbol{B}^*\boldsymbol{A}^*) \tag{12}$$

$$\sigma^2 = \mathbb{E}(y_{k,i} - \boldsymbol{x}_{k,i}^\top(\boldsymbol{W}_0 + \boldsymbol{B}^*\boldsymbol{A}^*))^2 \tag{13}$$

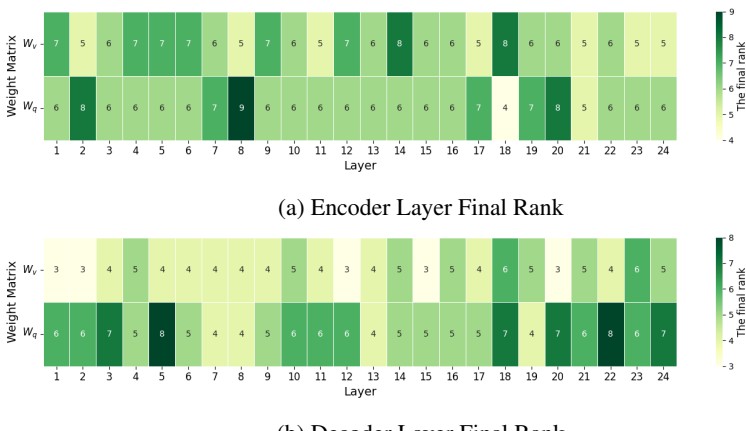

(a) Encoder Layer Final Rank

(b) Decoder Layer Final Rank

Figure 6: Comparison of the sum of the resulting ranks in the different modules of encoder and decoder Layers in CP-Rank training last three tasks of order 2 on standard CL benchmark. Here the $x$-axis is the layer index and the $y$-axis represents different modules (types) of low-rank weight matrices.

where $\sigma^2 > 0$ refers to the variance of the label noise.

**Assumption 4** (Commutable data covariance matrices). Assume that

$$\boldsymbol{H}_1 \boldsymbol{H}_2 = \boldsymbol{H}_2 \boldsymbol{H}_1, \quad \text{where } \boldsymbol{H}_1 = \frac{1}{n} \boldsymbol{X}_1^\top \boldsymbol{X}_1 \text{ and } \boldsymbol{H}_2 = \frac{1}{n} \boldsymbol{X}_2^\top \boldsymbol{X}_2 \tag{14}$$

Denote that due to low-rank nature and linear layers, the data representation can be represented by the low-rank parameter matrices $\boldsymbol{BA}$, thus we use the left singular vectors $\boldsymbol{U}$ to represent.

**Risk decomposition.** For simplicity, we use $\boldsymbol{\delta}_1 = \boldsymbol{B}_1 \boldsymbol{A}_1$, $\boldsymbol{\delta}_2 = \boldsymbol{B}_2 \boldsymbol{A}_2$, and $\boldsymbol{\delta}^* = \boldsymbol{B}^* \boldsymbol{A}^*$. According to the risk definition and the assumptions on the noise, we have

$$\begin{aligned}
\mathcal{R}_1(\boldsymbol{W}_0 + \boldsymbol{\delta}) &= \frac{1}{n_1} \mathbb{E} \|\boldsymbol{X}_1(\boldsymbol{W}_0 + \boldsymbol{\delta}_1) - y_1\|^2 \\
&= \frac{1}{n_1} \mathbb{E} \|\boldsymbol{X}_1(\boldsymbol{W}_0 + \boldsymbol{\delta}_1) - \boldsymbol{X}_1(\boldsymbol{W}_0 + \boldsymbol{\delta}^*) - \boldsymbol{\epsilon}_1\|^2 \\
&= (\boldsymbol{W}_0 + \boldsymbol{\delta}^*)^\top \boldsymbol{H}_1 (\boldsymbol{W}_0 + \boldsymbol{\delta}^*) + \sigma^2 \\
&= \langle \boldsymbol{H}_1, (\boldsymbol{W}_0 + \boldsymbol{\delta}^*)(\boldsymbol{W}_0 + \boldsymbol{\delta}^*)^\top \rangle + \sigma^2
\end{aligned} \tag{15}$$

Similarly, the risk for the second task is

$$\mathcal{R}_2(\boldsymbol{W}_0 + \boldsymbol{\delta}) = \langle \boldsymbol{H}_2, (\boldsymbol{W}_0 + \boldsymbol{\delta}^*)(\boldsymbol{W}_0 + \boldsymbol{\delta}^*)^\top \rangle + \sigma^2 \tag{16}$$

**Computing forgetting.** We now compute forgetting according to Eq. 15 and Eq. 16.

$$
\begin{aligned}
\mathcal{F} =& \mathbb{E}[\mathcal{R}_1(\boldsymbol{W}_0 + \boldsymbol{\delta}_1 + \boldsymbol{\delta}_2) - \mathcal{R}_1(\boldsymbol{W}_0 + \boldsymbol{\delta}_1)] \\
=& \langle \boldsymbol{H}_1, \mathbb{E}(\boldsymbol{W}_0 + \boldsymbol{\delta}_1 + \boldsymbol{\delta}_2 - (\boldsymbol{W}_0 + \boldsymbol{\delta}^*))(\boldsymbol{W}_0 + \boldsymbol{\delta}_1 + \boldsymbol{\delta}_2 - (\boldsymbol{W}_0 + \boldsymbol{\delta}^*))^\top \rangle \\
& - \langle \boldsymbol{H}_1, \mathbb{E}(\boldsymbol{W}_0 + \boldsymbol{\delta}_1 - (\boldsymbol{W}_0 + \boldsymbol{\delta}^*))(\boldsymbol{W}_0 + \boldsymbol{\delta}_1 - (\boldsymbol{W}_0 + \boldsymbol{\delta}^*))^\top \rangle \\
=& \langle \boldsymbol{H}_1, \mathbb{E}(\boldsymbol{\delta}_1 + \boldsymbol{\delta}_2 - \boldsymbol{\delta}^*)(\boldsymbol{\delta}_1 + \boldsymbol{\delta}_2 - \boldsymbol{\delta}^*)^\top \rangle - \langle \boldsymbol{H}_1, \mathbb{E}(\boldsymbol{\delta}_1 - \boldsymbol{\delta}^*)(\boldsymbol{\delta}_1 - \boldsymbol{\delta}^*)^\top \rangle \\
=& \langle \boldsymbol{H}_1, \mathbb{E}(\boldsymbol{\delta}_1 \boldsymbol{\delta}_1^\top + \boldsymbol{\delta}_1 \boldsymbol{\delta}_2^\top - \boldsymbol{\delta}_1 (\boldsymbol{\delta}^*)^\top + \boldsymbol{\delta}_2 \boldsymbol{\delta}_1^\top + \boldsymbol{\delta}_2 \boldsymbol{\delta}_2^\top - \boldsymbol{\delta}_2 (\boldsymbol{\delta}^*)^\top - \boldsymbol{\delta}^* \boldsymbol{\delta}_1^\top - \boldsymbol{\delta}^* \boldsymbol{\delta}_2^\top + \boldsymbol{\delta}^* (\boldsymbol{\delta}^*)^\top) \rangle \\
& - \langle \boldsymbol{H}_1, \mathbb{E}(\boldsymbol{\delta}_1 \boldsymbol{\delta}_1^\top - \boldsymbol{\delta}_1 (\boldsymbol{\delta}^*)^\top - \boldsymbol{\delta}^* \boldsymbol{\delta}_1^\top + \boldsymbol{\delta}^* (\boldsymbol{\delta}^*)^\top) \rangle \\
=& \langle \boldsymbol{H}_1, \mathbb{E}(\boldsymbol{\delta}_1 \boldsymbol{\delta}_2^\top + \boldsymbol{\delta}_2 \boldsymbol{\delta}_1^\top + \boldsymbol{\delta}_2 \boldsymbol{\delta}_2^\top - \boldsymbol{\delta}_2 (\boldsymbol{\delta}^*)^\top - \boldsymbol{\delta}^* \boldsymbol{\delta}_2^\top) \rangle \\
\overset{(1)}{=}& \operatorname{tr}(\boldsymbol{H}_1^T \mathbb{E}(\boldsymbol{\delta}_1 \boldsymbol{\delta}_2^\top + \boldsymbol{\delta}_2 \boldsymbol{\delta}_1^\top + \boldsymbol{\delta}_2 \boldsymbol{\delta}_2^\top - \boldsymbol{\delta}_2 (\boldsymbol{\delta}^*)^\top - \boldsymbol{\delta}^* \boldsymbol{\delta}_2^\top)) \\
\overset{(2)}{\leq}& \|\boldsymbol{H}_1\| \operatorname{tr}(\mathbb{E}(\boldsymbol{\delta}_1 \boldsymbol{\delta}_2^\top + \boldsymbol{\delta}_2 \boldsymbol{\delta}_1^\top + \boldsymbol{\delta}_2 \boldsymbol{\delta}_2^\top - \boldsymbol{\delta}_2 (\boldsymbol{\delta}^*)^\top - \boldsymbol{\delta}^* \boldsymbol{\delta}_2^\top)) \\
\overset{(3)}{=}& \|\boldsymbol{H}_1\| (2 \operatorname{tr}(\mathbb{E}\boldsymbol{\delta}_1 \boldsymbol{\delta}_2^\top) + \operatorname{tr}(\mathbb{E}\boldsymbol{\delta}_2 \boldsymbol{\delta}_2^\top) - 2 \operatorname{tr}(\mathbb{E}\boldsymbol{\delta}_2 (\boldsymbol{\delta}^*)^\top)) \\
\overset{(4)}{\leq}& \|\boldsymbol{H}_1\| (2 \operatorname{tr}(\mathbb{E}\boldsymbol{\delta}_1 \boldsymbol{\delta}_2^\top) + r \|\mathbb{E}\boldsymbol{\delta}_2\|_2^2 - 2 \operatorname{tr}(\mathbb{E}\boldsymbol{\delta}_2 (\boldsymbol{\delta}^*)^\top))
\end{aligned}
\tag{17}
$$

where $r$ is the rank of $\boldsymbol{\delta}_2$. To explain the derivation steps clearly, we assume that $\boldsymbol{A}$ and $\boldsymbol{B}$ are two matrices, (1) is from $\langle \boldsymbol{A}, \boldsymbol{B} \rangle = \operatorname{tr}(\boldsymbol{B}^T \boldsymbol{A})$, (2) is from $\operatorname{tr}(\boldsymbol{A}^\top \boldsymbol{B}) \leq \|\boldsymbol{A}\| \operatorname{tr}(\boldsymbol{B})$, (3) is from $\operatorname{tr}(\boldsymbol{A} + \boldsymbol{B}) = \operatorname{tr}(\boldsymbol{A}) + \operatorname{tr}(\boldsymbol{B})$, (4) is obtained from $\operatorname{tr}(\boldsymbol{\delta}_2 \boldsymbol{\delta}_2^\top) \leq r \|\boldsymbol{\delta}_2\|_2^2$.

**Analysis of the upper bound of $\mathcal{F}$.**

- The matrix $\boldsymbol{H}_1$ is a data correlation matrix, and in practice, we cannot directly control its magnitude.
- The term $r \|\mathbb{E}\boldsymbol{\delta}_2\|_2^2$ becomes smaller when using a lower rank for $\boldsymbol{\delta}_2$, which helps in reducing the overall forgetting error.
- Additionally, $\operatorname{tr}(\mathbb{E}\boldsymbol{\delta}_2 (\boldsymbol{\delta}^*)^\top)$ is also expected to be small when using a low-rank approximation for $\boldsymbol{\delta}_2$.

Therefore, the upper bound on the forgetting error is dominated by the term involving $\boldsymbol{\delta}_1 \boldsymbol{\delta}_2^\top$, denoted by:

$$
\mathcal{F} \leq \mathcal{O}(\operatorname{tr}(\boldsymbol{\delta}_1 \boldsymbol{\delta}_2^\top))
\tag{18}
$$

