# OpenReview forum: "Combating the Generalization-Forgetting Trade-off in Continual Learning: A Cautious Passive Low-Rank Approach"
_ICLR.cc/2025/Conference — Submitted to ICLR 2025_

### Official Review · Reviewer_2i5r · 2024-10-25

**Soundness:** 3
**Presentation:** 3
**Contribution:** 3
**Rating:** 6
**Confidence:** 4

**Summary:**

This manuscript focused on continual learning (CL) with Large Language Models (LLM). The authors proposed a parameter-efficient approach based on the low-rank adaptation (LoRA) and explored the role of layerwise ranks in the incremental learning of LoRA between different tasks for CL. Through some empirical results, the authors observed that a trade-off between low ranks and high ranks can be leveraged to balance forgetting mitigation and generalization. Based on this motivation, the authors proposed Cautious Passive Low-Rank (CP-Rank) that gradually increases the rank of layerwise weight matrices during training. Specifically, the similarity of between-task low-rank subspaces is measured to evaluate the orthogonality between subspaces, and then whether to cautiously increase the ranks or passively maintain the current ranks can be determined for each task. Experiments on several benchmarks were conducted to support the effectiveness of the proposed method CP-Rank.

**Strengths:**

1. The motivation of this manuscript is clear. Determining how to increase the ranks is also novel for the continual learning for LLM with LoRA strategy.
2. Extensive experiments on different benchmarks were conducted to demonstrate the effectiveness. The datasets adopted in this manuscript include a wide range of task types.
3. This paper is well-written and easy to follow.

**Weaknesses:**

1. The mathematical presentation can be further improved. There are some confusing points that need to be clarified.
2. The paper’s focus is primarily on state-of-the-art low-rank methods, but including a few recent non-low-rank continual learning methods as baselines could provide a broader performance perspective.

**Questions:**

1. In Line 198, the sentence "Eq. 7 uses the singular values captured by two different task subspaces, which matches our findings in Eq. 5" is hard to follow. It would be better if the authors could provide intuitive explanations about the connection with Eq. 5.
2. In Eq. (2), I guess there is a typo for the subscript before the minus sign. Should it be $\mathbf{\theta}_N$ instead of $\mathbf{\theta}_T$? Besides, in Eq. (2) and Eq. (3), the parenthesis can be added for the different terms of RHS to avoid confusion.
3. In Eq. (7), I wonder if there are some typos about the indices (e.g., $i,s$). This equation is confusing.
4. In Algorithm 1, I noticed that an interval $k$ was set to control the operations within Step 4. However, I didn't see any description of this point. Could the authors explain the role of this interval $k$?
5. Some superscripts in Section 2 are a little messy. For example, in Line 177, the superscript is adopted to indicate the layer $l$. However, in Algorithm 1 and Algorithm 2, the matrices $\mathbf{A}$ and $\mathbf{B}$ were subscripted with a time step $t$. This confusing part needs to be further clarified.
6. I noticed that other studies, such as InfLoRA [1], also considered orthogonality between the subspaces of different tasks during the LoRA adaptation. Could the authors summarize the main differences between their proposed CP-Rank ad InfLoRA?

References;

[1] InfLoRA: Interference-Free Low-Rank Adaptation for Continual Learning. CVPR 2024.

---

> ### Author Response · Authors · 2024-11-26
>
> Thank you for your appreciation and excellent summary of our work. We also appreciate the time and effort you dedicated to reviewing our research. We have addressed your questions and concerns below:
>
>
> > **W1: About mathematical presentation**
>
> Thanks for your suggestion. We would clarify the mathematical presentation better and would improve it.
>
> > **W2: About non-low-rank continual learning**
>
> Thanks for your suggestion. We agree that including a few recent low-rank continual learning methods as baselines can provide a broader performance perspective. Since we focus on parameter-efficient continual learning in our work and consider low-rank methods severing as a core aspect, we didn't choose non-parameter-efficient continual learning methods as baselines. We plan to explore non-low-rank methods in the following to make our work have a more insightful performance analysis.
>
> > **Q1: About connection between Eq.7 and Eq.5**
>
> Thanks for your comment on this connection. We would like to explain: Eq.7 is defined as $\phi(U_{i}^{l},U_{j}^{l}) = \frac{\sum_{s=1}^{p}\sigma_{s}^2}{p} = \frac{1}{p}(1-d(U_{i}^{l},U_{j}^{l})^2)$, where $\sigma_{s}$ are the singular values of $(U_{i}^{l})^\top U_{j}^{l}$, and $U_{i}^{l}$ and $U_{j}^{l}$ are the left singular matrices of $(B_i^{l}A_i^{l})^\top B_i^{l}A_i^{l}$ and $B_j^{l}A_j^{l}$, respectively. The singular values of $B_j^{l}A_j^{l}$ depend on the alignment of left and right singular vectors. And $(U_{i}^{l})^\top U_{j}^{l}$ captures the cosine of the principal angles between subspaces spanned by $U_{i}^{l}$ and $U_{j}^{l}$. Thus, the singular values of $(U_{i}^{l})^\top U_{j}^{l}$ are proportional to singular values of $(B_i^{l}A_i^{l})^\top(B_j^{l}A_j^{l})$. So we have $\sum_{s=1}^{p}\sigma_{s}^2\propto\text{tr}((U_{i}^{l})^\top U_{j}^{l})\propto\text{tr}((B_i^{l}A_i^{l})^\top(B_j^{l}A_j^{l}))$.
> It indicates that singular values of $(B_i^{l}A_i^{l})(B_j^{l}A_j^{l})^\top$ in Eq.5 are connected to singular values of $(U_{i}^{l})^\top U_{j}^{l}$ in Eq.7. We will make detailed explanation in the revised manuscript.
>
>
> > **Q2: About typo in Eq.(2) and presentation of Eq.(2)(3)**
>
> Thanks for your correctness. Yes, it should be $\Theta_{N}$ instead of $\Theta_{T}$. We present the modified version of Eq.(2) and Eq.(3):
> $F(\theta_{1},\dots,\theta_{N}) = \sum_{t=1}^{N-1}(L_{t}(\theta_{N})-L_{t}(\theta_{t}))$ and
> $I(\theta_{1},\dots,\theta_{N}) = \sum_{t=1}^{N}(L_{t}(\theta_{t})-L_{t}(\theta_{t}^{*}))$
>
> > **Q3: About typo in Eq.(7)**
>
> Thanks for pointing out typos. Yes, there are typos in Eq.(7), and sorry for causing confusion. It should be
> $\phi(U_{i}^{l}, U_{j}^{l}) = \frac{\sum_{s=1}^{p}\sigma_{s}^2}{p} = \frac{1}{p}(1-d(U_{i}^{l},U_{j}^{l})^2)$
>
> > **Q4: About rank updating interval $k$**
>
> Thanks for your question. The rank updating interval $k$ is used to control the frequency of subspace similarity evaluation during training, which means when to invoke Algorithm 2 for rank update. We increase the rank (randomly initialize newly added matrix elements) in Algorithm 2 and these new elements require training for a while before the next subspace similarity evaluation. The interval $k$ is the number of iterations between each subspace similarity evaluation. We also evaluate the impact of $k$ in Figure 4 on Page 10.
>
>
> > **Q5: About superscripts in Section 2**
>
> Thank you for your suggestion. We agree that the superscripts in Section 2 should be clearer. In Algorithm 2, $A_{i}^{t}$ represents task $T_{i}$ and the current iteration number $t$, which is different from $A_{i}^{l}$ we use to denote the $l$-th layer. To avoid further confusion, we chose not to use $A_{i}^{l,t}$, as it could make the notation overly complex. Instead, as noted in line 201 "For simplicity, we use $B$ and $A$ to represent the layer-wise weight matrices $B^{l}$ and $A^{l}$."
>
>
> > **Q6: About the difference between InfLoRA and CP-Rank**
>
> Thanks for highlighting this related work. We appreciate the opportunity to compare our method with it.
>
> - **Model**: InfLoRA proposes a LoRA-based continual learning method for a pre-trained Vision Transformer (ViT), while our approach, CP-Rank, is for general large language models and not a specific LLM.
>
> - **Method**: 1. InfLoRA uses fixed and same rank for each LoRA without considering rank patterns of tasks and model different layers, while CP-Rank considers rank patterns among tasks and model layers. 2. InfLoRA not only stores previous LoRA parameters but also additionally stores gradient subspaces of previous tasks, while CP-Rank initializes new task subspaces without storing extra components, which is more memory-efficient.
>
> - **Dataset**: InfLoRA uses image datasets, while CP-Rank is applied to text (natural language) datasets.
>
> InfLoRA provides CL method to vision-language models, while our method offers a scalable and effective approach to CL in LLMs, proven in NLP tasks. We are committed to including this related work and comparison in the subsequent revision.

---

> > ### Comment · Reviewer_2i5r · 2024-11-28
> > **Further Questions on Author's Rebuttal**
> >
> > Thanks for providing explanations to my questions. After carefully reading it, I have some further questions:
> >
> > 1. I read the proof of Eq. (5) and I don't think the process of obtaining this equation is smooth. I checked Appendix A.5 and I have to say that the derivation of obtaining this equation is really not rigorous. The authors only analyzed the simple "linear regression" case, which is very toy and I have relevant concerns about the huge gap between the theoretical analysis and practical algorithm.
> >
> > 2. In your previous answer to my original Question (1), I wonder why $U_{i}^{l}$ is the left singular matrices of $(B_{i}^{l} A_{i}^{l})^\top B_{i}^{l} A_{i}^{l}$ ...
> >
> > 3. About the comparisons with InfLoRA. I cannot agree with some points: (1) I don't think InfLoRA can only be applied to pre-trained ViT and it can also be adapted to other architectures; (2) InfLoRA is not tailored for vision-language models. Thus, I believe the authors should provide empirical comparisons with InfLoRA, e.g., applying InfLoRA to your datasets and compare the performances.

---

> > > ### Author Response · Authors · 2024-12-01
> > >
> > > Thank you for your detailed feedback and for the opportunity to address your concerns. We value your insights and have provided responses to your key questions below.
> > >
> > > > **Q1: About Eq.5**
> > >
> > > Thank you for carefully reviewing our appendix. We agree that the derivation of Eq.5 is a toy, as stated in our manuscript (line 946 on page 18). Its primary purpose is to provide a high-level intuition rather than a rigorous theoretical analysis, which we see as an interesting avenue for future work. The theoretical understanding of continual learning, particularly for LLMs, remains underdeveloped, focusing mainly on two-task settings in simple linear models, leaving the complexities of continual learning across multiple tasks largely unexplored. Our work prioritizes an empirical algorithm and illustrates the connection through a basic case.
> > >
> > > Our experimental results show that CP-Rank effectively improves the performance of continual learning for LLMs. We hope this explanation clarifies the intention behind the derivation and emphasizes the empirical contributions of our work.
> > >
> > > > **Q2: About the answer to original Q1**
> > >
> > > Thanks for pointing it out. $U_{i}^{l}$ is the left singular matrix of $B_{i}^{l} A_{i}^{l}$. This can be found in our original manuscript (line 179-line 180). We sincerely apologize for the typo (manual mistake) in our previous response in this window and truly appreciate your understanding.
> > >
> > >
> > > > **Q3: About the comparison with InfLoRA**
> > >
> > > Thank you for your thoughtful discussion of InfLoRA. We appreciate the opportunity to further engage with this work. In Algorithm 1 in the paper of InfLoRA (page 5), they assume that $f_{\Theta}(\cdot)$ is a pre-trained ViT model. Furthermore, in their experimental setup described on page 6, the authors specify the use of the ViT-B/16 backbone. Upon carefully reviewing all the citations of the InfLoRA paper, we found that all citing works are from the computer vision field, using ViT-based models and image-related datasets for their experiments. In these cases, InfLoRA is used as a baseline. Similarly, O-LoRA is not used as a baseline in these works, as it is focused on NLP tasks.
> > >
> > > Additionally, we carefully examined the framework of InfLoRA and believe that it may be adaptable to general LLMs, as its design does not inherently rely on ViT-specific elements. However, the publicly available implementation code of InfLoRA is specific to ViT models and image datasets, thus it will take time for us to implement it on our own. We understand the importance of using InfLoRA as a baseline to provide a comprehensive comparison. If you consider it crucial to evaluate InfLoRA in the NLP field, we are fully committed to implementing it and including a detailed comparison in our revised manuscript. We appreciate your understanding and look forward to addressing this aspect in our further work.

---

### Official Review · Reviewer_TV91 · 2024-10-25

**Soundness:** 2
**Presentation:** 2
**Contribution:** 3
**Rating:** 3
**Confidence:** 4

**Summary:**

This paper proposes CP-Rank (Cautious Passive Low-Rank), a continual learning strategy for LLM to balance the trade-off between low ranks and high ranks to balance forgetting mitigation and generalization based on LORA. It is observed that updating a high-rank weight on new task is likely to contribute to better performance on the new-task but greater forgetting previous tasks, and vice versa. CP-Rank progressively and adaptively increase the rank of derivative weight matrix the new task according to its similarity with previous tasks'.
Experiments show that it is effective.

**Strengths:**

The method is intuitive and basically reasonable. It is shown that the method could result in significant advantage in empirical performance.

**Weaknesses:**

1. The most concerned weakness is the efficiency problem of CP-Rank. In terms of space complexity, it is required to memorize the derivative weight matrix on every layers of every task, which takes NLd^2. In terms of time complexity, to learn a single new task, there are additionally T/k times to refer to all the NL matrix in memory and respectively perform SVD and evaluate the Grassmann similarity. It is like to be very inefficient in both space and time.
2. The very first experiment that gives rise to the fundamental question is not convincing enough. The results on only one pair of tasks are shown, which is far not enough to support the observed pattern statistically. Though it is intuitive that low-rank updating has advantage in unforgetting and disadvantage in new new loss compared with high-rank.
3. Eqn(5) dos not match the method design well. There are obvious gap between Eqn(5) and the method design in terms of problem setting and similarity metric.
4. There are some redundant contents and detail errors that trouble reading. For example, it seems that eqn(2)(3)(4) is very unnecessary; The $T$ in eqn(2) is likely to be correct to be $N$; Eqn(7) is not likely to has a codomain $(0,1)$.

**Questions:**

There is one main question that the reviewer is curious about and would appreciate to discuss with the authors: Is there a correlation between the rank of the derivative matrix of the new task and its mean similarity with previous tasks? It is demonstrated that seems similarity and rank are two independent factors. CP-Rank progressively increase the rank under a similarity threshold, which means the similarity corresponding to unforgetting, is prior to the rank corresponding to better fitting. This is reasonable from a practical perspective to just adding untrained rows and columns to AB while not disturbing the existing rows and columns. But it is likely that there is such a correlation which can be exploit.

For example, a positive correlation can guide the early-stopping during increasing the rank to be more efficient.

---

> ### Author Response · Authors · 2024-11-26
>
> Thank you for your valuable feedback on our work. We truly appreciate the time you invested in the review. We have carefully considered your insights and addressed the highlighted concerns. We hope our responses provide clarity on the matters raised.
>
> > **W1: About the efficiency of CP-Rank**
>
> Thanks for your question about the efficiency of CP-Rank.
>
> - **Space complexity**: We reduce the memory by using low-rank matrices and the memory should be $NL(d_1\times r+d_2\times r)$, where $B\in\mathbb{R}^{d_1\times r}$ and $A\in\mathbb{R}^{r\times d_2}$, where $r\ll\text{min}(d_1,d_2)$. Compared to full fine-tuning, we only need to train LoRA parameters, significantly saving memory while achieving comparable performance. Additionally, we use layerwise ranks for LoRA in each layer of each task, meaning ranks may vary across layers within a task.
>
> - **Time complexity**: Performing SVD is necessary to obtain the representation of a low-rank subspace, a common method in low-rank works [1, 2]. Besides, our method randomly initializes new components of LoRAs during training, meaning $A$ or $B$ alone cannot represent low-rank subspaces. Thus, to ensure correct representation of low-rank subspaces, we obtain the left singular matrix via SVD, which is then used to calculate subspace similarity.
>
> We will make these complexities clear in the revised version.
>
> > **W2: About the first experiment**
>
> Thank you for pointing this out. This experiment serves as an example to empirically show that there exists a trade-off between low and high ranks in balancing forgetting mitigation and generalization amplification. It does not imply a straightforward linear relationship between rank size and the degree of forgetting. The motivation for our work comes from several studies [3,4] which have shown that LoRA forgets less than more regularization techniques like weight decay and dropout and also helps maintain the diversity of generations, while other studies [5,6] indicate that low-rank update mechanism limits LLMs' ability to learn as effectively as full fine-tuning, and using higher-rank update mechanism can help LoRAs achieve better performance.
>
> > **W3: About Eqn(5)**
>
> Thanks for raising this comment. Eqn(5) motivates the effectiveness of Eqn(7), which measures the similarity of low-rank subspaces. Eqn(7) is defined as $\phi(U_{i}^{l},U_{j}^{l}) = \frac{\sum_{s=1}^{p}\sigma_{s}^2}{p} = \frac{1}{p}((1-d(U_{i}^{l},U_{j}^{l})^2)$, where $\sigma_{s}$ are the singular values of $(U_{i}^{l})^\top U_{j}^{l}$, and $U_{i}^{l}$ and $U_{j}^{l}$ are the left singular matrices of $(B_i^{l}A_i^{l})^\top B_i^{l}A_i^{l}$ and $B_j^{l}A_j^{l}$, respectively. The singular values of $B_j^{l}A_j^{l}$ depend on the alignment of left and right singular vectors. And $(U_{i}^{l})^\top U_{j}^{l}$ captures the cosine of the principal angles between subspaces spanned by $U_{i}^{l}$ and $U_{j}^{l}$. Hence, the singular values of $(U_{i}^{l})^\top U_{j}^{l}$ are proportional to the singular values of $(B_i^{l}A_i^{l})^\top(B_j^{l}A_j^{l})$. Thus, this relationship is expressed as $\sum_{s=1}^{p}\sigma_{s}^2\propto\text{tr}((U_{i}^{l})^\top U_{j}^{l})\propto\text{tr}((B_i^{l}A_i^{l})^\top(B_j^{l}A_j^{l}))$, showing that the singular values of $(B_i^{l}A_i^{l})(B_j^{l}A_j^{l})^\top$ in Eqn(5) are connected to the singular values of $(U_{i}^{l})^\top U_{j}^{l}$ in Eqn(7). We will provide detailed explanations in the revised manuscript.
>
> > **W4: About explanation of Eqn(2)(3)(4)(7)**
>
> Thanks for your careful reading and correctness. For Eqn(2)(3)(4), we would like to show that our work aims to minimize the forgetting error via a well-designed update in the algorithm. We connect Eqn(5), the forgetting error bound, with Eqn(4), showing the impact of the forgetting error in continual learning optimization problem. For Eqn(2), the correct notation should be $N$. For Eqn(7), the codomain of Grassmann distance is $[0,\sqrt{p}]$, as mentioned in [7]. We use a reverse metric of the standard Projection Metric of Grassmann Distance, and its codomain is $[0,1]$ when the codomain of Grassmann distance is $[0,\sqrt{p}]$, as noted in [8].
>
>
> > **Q: About correlation between new task's rank and subspace similarity of tasks**
>
>
> Thanks for your interest in this relationship. We use the reversed Grassmann similarity to measure the similarity between low-rank subspaces of tasks for determining if low-rank subspaces of two tasks are orthogonal. If the subspace similarity is below the threshold, we assume subspaces are orthogonal, meaning tasks do not interfere with each other. Here, as noted in our responses to Weakness2, low-rank updates may limit LLMs ability. Thus, when tasks' subspaces are orthogonal, we consider this orthogonality to give new task a chance for higher rank updates during training to improve performance. This is the reason for connecting rank-increasing updates with subspace similarity in our work, rather than connecting rank patterns with subspace similarity.

---

> > ### Author Response · Authors · 2024-11-26
> >
> > [1] Qingru Zhang, Minshuo Chen, Alexander Bukharin, Pengcheng He, Yu Cheng, Weizhu Chen, and Tuo Zhao. Adaptive budget allocation for parameter-efficient fine-tuning. In The Eleventh International Conference on Learning Representations, 2023.
> >
> > [2] Jiawei Zhao, Zhenyu Zhang, Beidi Chen, Zhangyang Wang, Anima Anandkumar, and Yuandong Tian. Galore: Memory-efficient LLM training by gradient low-rank projection. In 5th Workshop on practical ML for limited/low resource settings, 2024.
> >
> > [3] Dan Biderman, Jose Gonzalez Ortiz, Jacob Portes, Mansheej Paul, Philip Greengard, Connor Jennings, Daniel King, Sam Havens, Vitaliy Chiley, Jonathan Frankle, et al. Lora learns less and forgets less. arXiv preprint arXiv:2405.09673, 2024.
> >
> > [4] Rakib Hyder, Ken Shao, Boyu Hou, Panos Markopoulos, Ashley Prater-Bennette, and Salman Asif. Continual learning via low-rank network updates, 2022.
> >
> > [5] Yongchang Hao, Yanshuai Cao, and Lili Mou. Flora: Low-rank adapters are secretly gradient compressors. In Forty-first International Conference on Machine Learning, 2024.
> >
> > [6] Wenhan Xia, Chengwei Qin, and Elad Hazan. Chain of loRA: Efficient fine-tuning of language models via residual learning. In ICML 2024 Workshop on LLMs and Cognition, 2024.
> >
> > [7] Jihun Hamm and Daniel D Lee. Grassmann discriminant analysis: a unifying view on subspace-based learning. In Proceedings of the 25th International Conference on Machine Learning, pp. 376–383, 2008.
> >
> > [8] Edward J Hu, yelong shen, Phillip Wallis, Zeyuan Allen-Zhu, Yuanzhi Li, Shean Wang, Lu Wang, and Weizhu Chen. LoRA: Low-rank adaptation of large language models. In International Conference on Learning Representations, 2022.

---

> > > ### Comment · Reviewer_TV91 · 2024-11-28
> > > **Thanks for your response**
> > >
> > > Thanks for the authors' response. This work's research topic is interesting. However, considering the inherent complexity and the method's efficiency is related to model size, I think it is too inefficient to be practical, especially for LLMs. I retain my negative rating.

---

> > > > ### Author Response · Authors · 2024-12-01
> > > >
> > > > Thank you for your feedback and for acknowledging the interesting nature of our research topic.
> > > >
> > > > > **Response to the practical application**
> > > >
> > > > We appreciate your concerns regarding the practical application and efficiency of our method, particularly for LLMs. We would like to address this by sharing additional experimental results conducted on the Llama-2-7b-chat model using six NVIDIA A100 GPUs during the rebuttal discussion. These results demonstrate that our method is indeed practical for larger LLMs, as shown below:
> > > >
> > > > | **Method**  | **Order 1** | **Order 2** | **Order 3** |
> > > > |-------------|-------------|-------------|-------------|
> > > > | O-LoRA      | 76.1    | 76.3   | 76.4    |
> > > > | **CP-Rank**     | **77.5**    | **77.6**    | **77.3**    |
> > > >
> > > >
> > > > These experiments highlight that our method (CP-Rank) achieves competitive performance across multiple task orders while being feasible to implement on large models like Llama-2-7b-chat. We kindly request you to consider these results, which we believe address concerns regarding the practicality and scalability of our approach for LLMs. Your feedback is valuable, and we hope this additional data clarifies the applicability of our method. We would be happy to relieve any further concerns you might have.

---

### Official Review · Reviewer_BR1X · 2024-10-27

**Soundness:** 2
**Presentation:** 3
**Contribution:** 2
**Rating:** 5
**Confidence:** 4

**Summary:**

This paper presents CP-Rank, a parameter-efficient continual learning method that progressively increases the layer-wise rank of LoRAs for new tasks, guided by a low-rank subspace similarity metric between tasks. CP-Rank not only models the low-rank relationships between tasks' incremental LoRAs but also adapts to the unique low-rank dynamics across different layers of the model. This approach effectively prevents forgetting of previous tasks while improving generalization on new tasks, all within a memory-efficient framework. Extensive experiments on natural language processing and complex math reasoning tasks demonstrate that CP-Rank effectively captures rank patterns in continual learning and consistently outperforms existing approaches.

**Strengths:**

* This paper is easy to follow.

* The proposed approach is simple and easy to understand.

**Weaknesses:**

* The method would benefit from evaluation on a larger language model, as T5-large may not be sufficiently large for comprehensive assessment.

* The approach for increasing the LoRA rank appears heuristic and lacks sufficient justification. A deeper explanation and analysis of the rank-increasing strategy would strengthen the paper.

* The proposed method introduces several hyperparameters (e.g., $k$, $\epsilon$, and the rank-increasing mechanism), which could complicate its practical application. A simpler or more streamlined approach might improve usability.

* The relationship between low rank and forgetting, and high rank and generalization, is not well-justified. While the introduction offers empirical evidence of this correlation, a formal explanation or theoretical foundation is missing. Additionally, there is no clear discussion of whether the method generalizes to larger models, such as Llama.

* Other:
The font size in Figure 1 is too small, making it difficult to read. Increasing the size would improve clarity.

**Questions:**

N/A

---

> ### Author Response · Authors · 2024-11-26
>
> Thank you for your valuable and helpful feedback on our work. We have carefully considered your concerns, and we hope that our responses provide clear explanations for the problems raised.
>
> > **W1 and W4: About experiments on larger language models**
>
> Thanks for your suggestion. We use T5-large to show main results in our manuscript since O-LoRA uses T5-large to present. Here, to show the effectiveness of our method on larger LLMs, we present the experimental results using Llama-2-7b-chat model to show the performance of CP-Rank with orthogonal projection. We use 6 NVIDIA A100 GPUs to conduct the following experiments on three task orders in standard CL benchmark.
>
>
> | **Method**  | **Order 1** | **Order 2** | **Order 3** |
> |-------------|-------------|-------------|-------------|
> | O-LoRA      | 76.1        | 76.3        | 76.4        |
> | CP-Rank     | 77.5        | 77.6        | 77.3        |
>
> These results indicate that our method can also work on larger LLMs. We are committed to using Llama-2-7b-chat or larger models in the revised manuscript.
>
>
> > **W2: About explanation and analysis of rank-increasing strategy**
>
> Thanks for pointing out this point. The method of increasing ranks during training has been studied and well-supported in existing works. In theory, greedy low-rank learning [1] characterizes the trajectory of stochastic gradient descent, which performs a rank-constrained optimization and greedily increases the rank whenever it fails to reach a global minimizer. In practice, [2] proposes the InRank algorithm which increases the rank during training to find intrinsic low rank of networks. These studies show that increasing the rank during training helps models find optimized low-rank solutions. Our work differs from the above works as we focus on continual learning on a sequence of tasks, rather than training on a single task. We adapt the rank-increasing method to help LLMs find low-rank matrices that satisfy continual learning metrics for each task.
>
> [1] Zhiyuan Li, Yuping Luo, and Kaifeng Lyu. "Towards resolving the implicit bias of gradient descent for matrix factorization: Greedy low-rank learning." arXiv preprint arXiv:2012.09839, 2020.
>
> [2] Jiawei Zhao, Yifei Zhang, Beidi Chen, Florian Tobias Schaefer, and Anima Anandkumar. Incremental low-rank learning. In Workshop on Efficient Systems for Foundation Models @ ICML2023, 2023.
>
> > **W3: About hyperparameters**
>
> Thanks for your interest in practical application. In our approach, the only hyperparameters are the updating interval $k$ and subspace similarity threshold $\epsilon$. The rank-increasing mechanism increases rank by 1 when the subspace similarity satisfies the threshold $\epsilon$, without requiring additional hyperparameters. We consider $k$ and $\epsilon$ as essential conditions that cannot be omitted in the algorithm. Threshold $\epsilon$ is inherently required in rank-increasing algorithms, such as [1], to determine when to increase the rank. Updating interval $k$ is crucial to control the frequency of subspace similarity evaluations. Calculating the subspace similarity at each training step would be computationally expensive. Moreover, $k$ ensures that newly added matrix elements are sufficiently trained before the next similarity evaluation. Thus, $k$ and $\epsilon$ are indispensable for ensuring the effectiveness of our method.
>
>
> [1] Jiawei Zhao, Yifei Zhang, Beidi Chen, Florian Tobias Schaefer, and Anima Anandkumar. Incremental low-rank learning. In Workshop on Efficient Systems for Foundation Models @ ICML2023, 2023.
>
>
> > **W4: About relationship between low rank and forgetting, and high rank and generalization.**
>
> Thanks for your comment. We agree that a solid theory for this relationship would significantly enhance our work. However, to the best of our knowledge, we are the first to explore and apply this relationship in the context of continual learning for LLMs. Our primary objective is to investigate whether this empirical relationship can effectively improve parameter-efficient continual learning in LLMs. To this end, we propose CP-Rank, an empirical method that dynamically increases the rank for new tasks based on subspace similarity conditions. It is also worth noting that the theoretical understanding of continual learning for LLMs remains underdeveloped, even the theory of continual learning primarily focuses on two-task settings, leaving complexities of continual learning across multiple tasks largely unexplored. Our work is to show that the empirical relationship between low rank and forgetting, and high rank and generalization, can be effectively leveraged to enhance continual learning performance. We believe that establishing a theoretical foundation for this relationship is a highly promising direction for future research.
>
>
> > **W5: About Figure 1**
>
> Thanks for your suggestion. We will increase the font size of Figure 1 to make it clear enough for readers in the revised manuscript.

---

> > ### Comment · Reviewer_BR1X · 2024-12-02
> >
> > Thanks for the rebuttal, the rebuttal addressed some concerns, but the method itself needs to be improved with a more solid foundation. I maintain my score.

---

### Official Review · Reviewer_HrN9 · 2024-10-29

**Soundness:** 3
**Presentation:** 2
**Contribution:** 3
**Rating:** 5
**Confidence:** 5

**Summary:**

This paper presents a continual learning method called CP-RANK, which achieves a balance between mitigating catastrophic forgetting and enhancing generalization by adjusting the rank across different layers. CP-RANK assigns each task a LoRA module as its adapter, using SVD decomposition to obtain the left singular matrix to assess the similarity between task subspaces. By comparing this similarity to a threshold, it determines whether the tasks are orthogonal. If they are orthogonal, the rank is increased; otherwise, the rank is maintained. Additionally, the paper introduces an improved approach for orthogonal projection. Comparative experiments are conducted to demonstrate the effectiveness of this method, and a necessary discussion on the experiments is provided.

**Strengths:**

- This paper proposes that low ranks are beneficial for resisting catastrophic forgetting, while high ranks are advantageous for learning new knowledge, with experiments conducted to support this perspective.
- The paper introduces a continual learning method called CP-RANK, which assigns different ranks to matrices at different layers and leverages rank increase to achieve a balance between mitigating catastrophic forgetting and enhancing generalization. Notably, the method incorporates an orthogonal projection algorithm, yielding promising results in experiments.
- It introduces the use of the left singular matrix to calculate the similarity between different subspaces to assess their orthogonality.
- The experimental section specifically evaluates performance on mathematical tasks.
- The paper provides detailed algorithms and formulas relevant to the proposed methods.

**Weaknesses:**

- The experimental results indicate that CP-RANK without orthogonal decomposition performs worse than O-LoRA, and in testing its resistance to catastrophic forgetting, the original CP-RANK setup was not included. This may be insufficient to demonstrate the effectiveness of rank adjustments for continual learning, as it still primarily relies on the orthogonal projection method used by O-LoRA.
- In Algorithm 2, for situations where rank increase is needed, the newly added matrix elements are initialized randomly. I find the explanation for this part of the algorithm unclear: after increasing the rank, is additional training required? Should orthogonality be reassessed?
- I believe that using "plasticity" (the capacity to learn new knowledge) in place of "generalization" might be more appropriate in this context, as the paper does not include experiments specifically validating generalization.

**Questions:**

- The paper claims that a major advantage of this model is that it can operate without relying on task IDs. I would like to understand how the model achieves task recognition and adapter selection in this case.

---

> ### Author Response · Authors · 2024-11-26
>
> Thank you for your detailed feedback on our work. We have thoughtfully addressed your insights and concerns, and hope our responses offer the necessary clarity on the issues raised.
>
> > **W1: About CP-Rank without orthogonal projection**
>
> Thanks for your question about the CP-Rank without orthogonal projection. The only difference between CP-Rank with/without orthogonal projection is the loss function. CP-Rank without orthogonal projection exactly achieved Algorithm 1 and Algorithm 2 with the loss in Eqn(1), without any orthogonal regularizer. CP-Rank (wo OP) performance only depends on the rank-increasing update. In Table 1 of our manuscript, CP-Rank (wo OP) outperforms other baselines except O-LoRA, which means only using the well-designed rank-increasing update without any explicit orthogonal methods during training can overcome forgetting and achieve generalization. Thus, CP-Rank (wo OP) reveals that leveraging low-rank property during updating can balance the trade-off in continual learning. In contrast, CP-Rank (w OP) uses the loss with the orthogonal regularizer to make subspaces of the new task gradually orthogonal to the subspaces of previous tasks, and with the help of orthogonality, CP-Rank (w OP) would perform better than CP-Rank (wo OP). To understand the effectiveness of rank-increasing update in CP-Rank (wo OP), we compare it with IncLoRA (uses a fixed rank of 8 for each new task with loss in Eqn(1)):
>
> | **Method**            | **Order 1** | **Order 2** | **Order 3** |
> |------------------------|-------------|-------------|-------------|
> | IncLoRA               | 63.4        | 62.2        | 65.1        |
> | CP-Rank (wo OP)       | 72.8        | 73.7        | 70.7        |
>
>
> Across all orders, CP-Rank (wo OP) outperforms IncLoRA by about 10\%. This improvement is only caused by the well-designed rank-increasing update, which means only dedicatedly increasing ranks when subspaces of tasks are assumed to be orthogonal can improve the performance greatly.
>
>
> > **W2: About newly added matrix elements in Algorithm 2**
>
> Thanks for your interest in newly added matrix elements. The newly added matrix elements are randomly initialized when the rank is increased and are trained alongside the original elements in subsequent iterations, which means they are not required to be retrained in the current iteration. In CP-Rank, the new subspace similarity would be calculated in the following iterations, including the newly added matrix elements which are treated as original elements in the following steps after rank increase. The orthogonality assessment would be in the following iterations rather than the current iteration.
>
> > **W3: About "plasticity" and "generalization"**
>
> Thanks for suggesting "plasticity". Since "generalization" also has the meaning of "the model's capacity to perform well on new, unseen data beyond the training set", we consider both "plasticity" and "generalization" to be used in learning new knowledge/tasks.
>
> > **Q: About testing without task IDs**
>
> Thanks for raising this question. In our continual learning setup, we use a newly initialized LoRA for each new task to train while freezing the pre-trained model parameters and previous tasks' LoRAs during the new task training. Previous tasks' LoRAs can be treated as concatenated with pre-trained model parameters. When training the new task, previous tasks' LoRAs and the pre-trained model parameters involve the inference (forward pass) even though they are frozen. When testing tasks, we use the entire model-including the pre-trained model, previous tasks' LoRAs, and new task LoRA-to test, which reflects the model's continual learning capabilities, as it measures performance after learning new tasks. Thus, in our setting and framework, there is no task recognition and adapter selection in testing.

---

### Official Review · Reviewer_zr9c · 2024-10-30

**Soundness:** 2
**Presentation:** 2
**Contribution:** 2
**Rating:** 3
**Confidence:** 5

**Summary:**

This work addresses the problem of continual learning for large language models (LLMs) and designs a mechanism to gradually increase the rank for the continual fine-tuning of LLMs.

**Strengths:**

The problem considered in this paper is meaningful in continual learning, and the parameter-efficient fine-tuning method employed is also a mainstream research approach currently.

**Weaknesses:**

1. Using only T5-large is insufficient to verify the effectiveness of the model on large language models (LLMs). Specifically, this paper claims to address the issue of continual fine-tuning in large models. However, the largest model used is only T5-large, which has only 770M parameters, far fewer than many existing LLMs such as Llama-2-7b, Llama-2-13b. The author should follow existing continual learning works [1, 2] that consider LLMs and perform experiments using Llama-2-7b and Llama-2-13b.

2. According to Equation 5, the motivation of this work is to keep $tr(B_{i}A_{i}(B_{j}A_{j})^{T})$ for ($j<i$) as small as possible when learning the i-th task. However, the algorithm ultimately aims to keep Equation 7 as small as possible. Equation 7 and $tr(B_{i}A_{i}(B_{j}A_{j})^{T})$ are not equivalent, leading to a mismatch between the motivation and the algorithm in this paper.

3. How is the second regularization term in Equation 8 derived? Because the lora branch $B_{i}A_{i}$ needs to be gradually updated during the learning of the $i$-th task, each calculation of the regularization term in Equation 8 requires SVD decomposition of $B_{i}A_{i}$, which will incur significant computational overhead.

[1] Zhao W, Wang S, Hu Y, et al. Sapt: A shared attention framework for parameter-efficient continual learning of large language models. Proceedings of the 62nd Annual Meeting of the Association for Computational Linguistics (Volume 1: Long Papers). 2024: 11641-11661.

[2] Wang X, Chen T, Ge Q, et al. Orthogonal subspace learning for language model continual learning[J]. arXiv preprint arXiv:2310.14152, 2023.

**Questions:**

Since the motivation of this algorithm is to maintain the orthogonality of $tr(B_{i}A_{i})$ and the lora branches of old tasks when learning a new task $i$, it is unreasonable to use a Gaussian distribution for random initialization of $B_{i}$. In other words, why not adopt some operations to maintain the orthogonality of $B_{i}A_{i}$ and the lora branches of old tasks during the initialization of $A_{i}$ and $B_{i}$?

---

> ### Author Response · Authors · 2024-11-26
>
> We sincerely appreciate your thorough review as well as constructive feedback. Your comments are extremely helpful. We have carefully addressed each of your concerns and provided detailed answers to your questions below:
>
> > **W1: About experiments using larger models**
>
> Thank you for suggesting the application of our method to larger models. In SAPT[1] and O-LoRA[2], the main results (results on continual learning benchmarks) shown in their paper both use T5-large model to present. Per your suggestion, to show the effectiveness of our method on larger language models, we would like to present the experimental results using Llama-2-7b-chat model to evaluate the performance of CP-Rank with orthogonal projection. We use 6 NVIDIA A100 GPUs to conduct the main experiments on three task orders in the standard CL benchmark.
>
>
> | **Method**  | **Order 1** | **Order 2** | **Order 3** |
> |-------------|-------------|-------------|-------------|
> | O-LoRA      | 76.1        | 76.3        | 76.4        |
> | CP-Rank     | 77.5        | 77.6        | 77.3        |
>
>
> These results indicate that our method can also work with larger language models. We are committed to using Llama-2-7b-chat or even larger models in the subsequent revision.
>
>
> > **W2: About connection between Equation 7 and Equation 5**
>
> Thank you for raising this comment. Equation 7 and Equation 5 are indeed related. Equation 7 is defined as:
> $\phi(U_{i}^{l},U_{j}^{l}) = \frac{\sum_{s=1}^{p}\sigma_{s}^2}{p} = \frac{1}{p}(1-d(U_{i}^{l},U_{j}^{l})^2)$, where $\sigma_{s}$ are the singular values of $(U_{i}^{l})^\top U_{j}^{l}$, and $U_{i}^{l}$ and $U_{j}^{l}$ are the left singular matrices of $(B_i^{l}A_i^{l})^\top B_i^{l}A_i^{l}$ and $B_j^{l}A_j^{l}$, respectively. The singular values of $B_j^{l}A_j^{l}$ depend on the alignment of their left and right singular vectors. And $(U_{i}^{l})^\top U_{j}^{l}$ captures the cosine of the principal angles between the subspaces spanned by $U_{i}^{l}$ and $U_{j}^{l}$. Consequently, the singular values of $(U_{i}^{l})^\top U_{j}^{l}$ are proportional to the singular values of $(B_i^{l}A_i^{l})^\top(B_j^{l}A_j^{l})$. This relationship can be expressed as: $\sum_{s=1}^{p}\sigma_{s}^2\propto\text{tr}((U_{i}^{l})^\top U_{j}^{l})\propto\text{tr}((B_i^{l}A_i^{l})^\top(B_j^{l}A_j^{l}))$. It shows that the singular values of $(B_i^{l}A_i^{l})(B_j^{l}A_j^{l})^\top$ in Equation 5 are connected to the singular values of $(U_{i}^{l})^\top U_{j}^{l}$ in Equation 7. We will provide a detailed explanation in the revised version of our manuscript.
>
> > **W3: About regularization in Equation 8**
>
> Thanks for your question about this. In our approach, we already perform SVD decomposition on LoRA $B_i A_i$ at the step when determining whether to increase the rank. During this step, we obtain the singular matrices, which can be directly used in Equation 8, avoiding redundant computations. In the "Rank Bonus" section, we introduce a regularization strategy similar to O-LoRA that includes a term $\sum_{i}A_i A_j$ to encourage orthogonality between the low-rank subspaces of different tasks. The goal of this regularization is to minimize interference across tasks by making their respective subspaces as orthogonal as possible. Furthermore, when increasing the rank, we initialize the new components randomly, and thus to effectively incorporate these new elements, we should update the loss function to include them. Since $B_i A_i$ can be spanned by its left singular matrix, we use this representation within the regularization term. This ensures the new task subspace spans a broader range, allowing the model to allocate a higher rank if needed, ultimately improving the performance of tasks.
>
>
> > **Q: About motivation of our algorithm**
>
> Thank you for suggesting methods to achieve orthogonality during the initialization of the new task LoRA components. In our algorithm, the primary objective is to dynamically determine when and how to increase the rank of the LoRA layers during training, rather than focusing on the initialization. Our approach leverages the low-rank property of LoRA to balance the trade-off between mitigating forgetting and enhancing generalization in continual learning. Specifically, we use the reversed Grassmann distance to measure the similarity between the subspace of the new task's LoRA and those of previous tasks. This metric helps assess subspace orthogonality and guides rank increases only when necessary. The reason we choose random initialization is that the goal of this work is to focus on the updating process in continual learning rather than the initialization. We consider designing an orthogonal initialization strategy as a promising direction for future research and plan to explore it further.

---

> > ### Comment · Reviewer_zr9c · 2024-11-28
> > **Thank you for the response**
> >
> > I believe there are some key points that need to be improved in this work.
> >
> > 1. In the experiments, it is necessary to use larger models to verify the effectiveness of the proposed method. Although the author has supplemented some experimental results using llama2-7b-chat, these result are still insufficient. Secondly, most of the experimental results presented by the author are on classification tasks. However, large models such as llama are capable of handling various types of natural language understanding and generation tasks. Therefore, I suggest that the author follow the existing work SAPT [1] and use the SuperNI benchmark to validate the effectiveness of the method. These experiments are unlikely to be completed during the rebuttal period.
> >
> > 2. Although the author has explained the connection between the method and its motivation in the rebuttal, I recommend that the author provide a mathematical proof, such as using a theorem to illustrate the connection between Equation 7 and Equation 5, and prove this theorem, which would be more convincing.
> >
> > Based on above points, I retain my rating.
> >
> > [1] Zhao W, Wang S, Hu Y, et al. Sapt: A shared attention framework for parameter-efficient continual learning of large language models. Proceedings of the 62nd Annual Meeting of the Association for Computational Linguistics (Volume 1: Long Papers). 2024: 11641-11661.

---

> > > ### Author Response · Authors · 2024-12-01
> > >
> > > Thank you for your thoughtful and constructive feedback on our work. We greatly appreciate the opportunity to address your points and clarify key aspects of our methodology.
> > >
> > > > **Point 1: About benchmarks in SAPT[1]**
> > >
> > > Thank you for suggesting the SuperNI benchmark. Our experiments share the same long-sequence tasks benchmark with SAPT (one of two benchmarks in their work), and we also incorporate two more challenging mathematical reasoning datasets (GSM8K and MATH), focusing on critical LLM capabilities.
> > >
> > > Besides, in continual learning, performance across different task orders is essential. We evaluate six task orders for standard and long-sequence benchmarks and two for mathematical datasets, compared to SAPT's four task orders, providing a broader assessment of continual learning.
> > >
> > > While we agree that including the SuperNI would strengthen our findings, we believe our benchmarks, particularly in mathematical reasoning, are more challenging. We are committed to incorporating SuperNI in future work and kindly request you consider these unique points.
> > >
> > > > **Point 2: About the explanation of the connection between Equation 7 and Equation 5**
> > >
> > > We agree that a rigorous proof would strengthen our work. However, the theoretical understanding of continual learning, especially for LLMs, remains underdeveloped, with existing theories focusing mainly on two-task settings. Our work contributes by providing high-level intuitions and empirical evidence rather than a fully developed theoretical framework, as noted in our manuscript. We view this as an exciting avenue for future work and look forward to further exploring these theoretical aspects to advance the field.

---

### Meta-Review · Area_Chair_3aLi · 2024-12-21

**Metareview:**

The paper presents CP-Rank, a parameter-efficient continual learning method for large language models that dynamically adjusts the rank of LoRA adapters to balance forgetting mitigation and generalization. Reviewers appreciate the novelty and intuitive nature of the approach but raised several significant concerns, including limited evaluations on larger language models besides T5-large (zr9c, BR1X), insufficient experiments and analysis (HrN9, BR1X, TV91), the mismatch between equations and the method’s design (TV91, zr9c, 2i5r), and the efficiency issues in algorithm design (TV91). While the authors made great efforts to address these concerns during the rebuttal phase, reviewers continued to express reservations about the experiment scales on the latest LLMs and theoretical foundations. Considering the reviewers’ opinions and the remaining concerns, the AC recommends further refining the paper before this paper can be accepted.

**Additional Comments On Reviewer Discussion:**

Overall, there are active discussions between the authors and some of the reviewers. The authors make great efforts to address the concerns of the reviewers. However, some of the concerns remain unresolved.


Reviewer Zr9c's major concern was the limited evaluation of the small model (T5-large), which is insufficient for verifying the effectiveness of larger LLMs like Llama-2-7b/13b. While the authors provided additional Llama-2-7b results, the reviewers noted the need for more comprehensive experiments.

Reviewer HrN9 questioned the effectiveness of CP-RANK without orthogonal projection, and the authors added corresponding experiment results during rebuttal. Without further engagement from the reviewer, the concerns are still carefully considered but less weighed.

Reviewer BR1X’s raised issues include the usage of small models in experiments, lacking theoretical justifications in the proposed approach, and hyperparameter choices. After the authors' rebuttal, the reviewer still expresses concerns regarding the theoretical foundation of the method.

Reviewer TV91 was mainly concerned about efficiency and computational complexity (frequent SVD operations and similarity calculations). The authors explained that using low-rank matrices reduces memory and that SVD operations are necessary for subspace representation, but given the impact of increasing model size, the reviewer remained unconvinced.

Reviewer 2i5r mentioned several confusions in the mathematical presentation and asked for clarifications of several concepts and baselines. After rebuttal, the reviewer further expressed concerns regarding the gap between the theoretical analysis and practical algorithm and suggested the need for further experiments of baselines.

Given the above unaddressed concerns and the reviewers' opinions, the AC recommends further refining the paper before this paper can be accepted.

---

### Decision · Program_Chairs · 2025-01-22

Reject